# Actively Enlarging Feature Norms with Universal Adversarial Training and Channel Calibration for Superior OOD Detection

## Abstract

Out-of-distribution (OOD) detection is an increasingly essential component for ensuring the safety and reliability of machine learning systems. A key insight in this area is that the feature norm gap between in-distribution (ID) and OOD data serves as a strong signal for identifying anomalous inputs. Building on this, we propose an innovative Universal Adversarial Training (UAT) framework that actively enlarges this norm gap. Our method introduces a single, learnable Universal Adversarial Map (UAM) that acts as a global regularizer to address shared vulnerable directions in the model. By regularizing the decision boundaries, this approach enlarges the gaps between ID classes, which enhances the model's generalization and its ability to distinguish OOD data. To further enhance detection at inference, we introduce a Channel-Calibrated Feature Norm (CCFN) scoring mechanism that refines the feature norm by suppressing irrelevant background activations. Our experiments and ablation studies demonstrate that these innovations lead to substantial performance gains across various benchmarks.

## 1 Introduction

While modern deep learning systems have achieved impressive performance across a wide range of tasks, they often struggle to recognize inputs that differ fundamentally from the data encountered during training. As AI technologies are increasingly deployed in real-world, high-risk environments, it has become essential to reliably identify out-of-distribution (OOD) inputs. OOD detection aims to distinguish data that belong to the training distribution (in-distribution, ID) from those that do not, thereby minimizing the risk of unintended or unsafe model behavior.

Early approaches to OOD detection focused on the model's output logits, introducing methods such as maximum softmax probability (MSP) (Hendrycks & Gimpel, 2016), ODIN with temperature scaling (Liang et al., 2017), and energy-based models (Liu et al., 2020). More recently, attention has shifted to the feature space, with studies showing that the L2 norm of feature vectors from the penultimate layer can provide a strong signal for distinguishing ID from OOD samples (Wang et al., 2022; Yu et al., 2023; Park et al., 2023). These works have demonstrated that ID features generally exhibit higher L2 norms than OOD features, often outperforming confidence scores derived from softmax outputs.

However, most feature norm-based methods passively rely on the naturally occurring gap between ID and OOD features, without actively promoting this separation during training. To address this, we introduce an innovative Universal Adversarial Training (UAT) framework that aims to actively and significantly enlarge the feature norm gap. Our approach is motivated by the need to address the *shared vulnerable directions* present in neural networks (Moosavi-Dezfooli et al., 2017), where small, consistent perturbations can cause misclassification across many samples. The core of our UAT is a learnable Universal Adversarial Map (UAM), which acts as a global regularizer that smooths the model's decision boundaries. This regularization process enlarges the gaps between different ID class distributions. Consequently, the model's generalization ability is enhanced. A model that is more adept at classifying known classes is inherently better at identifying inputs that do not belong to any of them. This improved discriminative power for ID data directly translates into a superior ability to detect OOD samples, as their features are less likely to fall within the now

well-defined class boundaries, resulting in lower feature norms. Furthermore, since the model is explicitly trained to recognize perturbed ID samples as familiar, while never seeing perturbed OOD samples, the UAM's effect on OOD data remains disruptive and novel, further amplifying the signal for detection.

Beyond training, we further enhance inference with our Channel-Calibrated Feature Norm (CCFN) mechanism. Traditional feature norm approaches aggregate all channels equally, which can allow irrelevant or background activations to mask important semantic features. CCFN mitigates this by recalibrating each feature channel according to its maximum activation, effectively suppressing background noise and highlighting informative regions, leading to more accurate OOD detection.

In summary, our contributions are twofold: first, we introduce a novel universal adversarial training framework and provide a rigorous theoretical foundation. We formally prove how UAT reshapes the feature space to systematically amplify the feature norm ratio between ID and OOD data. Second, we design the complementary channel-calibrated feature norm scoring mechanism to further enhance this separation at inference time. Together, these innovations yield substantial gains in OOD detection across diverse model architectures and benchmarks.

## 2 RELATED WORK

A variety of OOD detection methods have been proposed to handle distribution shifts at test time. These approaches are typically grouped into four main categories: those based on intrinsic model signals, augmented training data, open-world vision language models, and input perturbations.

**Based on Intrinsic Model Signals.** These methods rely on information produced by a pretrained model during inference to distinguish between ID and OOD samples. A classic line of work uses softmax-based confidence scores to separate ID from OOD data. Hendrycks & Gimpel (2016); Liu et al. (2020) make use of the model's softmax values to assign a confidence or an energy score, and then classify an input as OOD if the score crosses a chosen threshold. Other approaches, such as the Mahalanobis distance method (Lee et al., 2018), use feature representations to measure the distance between a sample and known class-conditional distributions. Going beyond output-level signals, researchers have noticed that OOD samples often produce unusually high activations in some neurons, which act as noise and can hinder OOD detection. Several approaches address this by clipping high activations, for example, in the penultimate layer, to reduce noise and improve feature discriminability (Sun et al., 2021; Sun & Li, 2022).

A significant area of research focuses on feature norm. Fang et al. (2022) established the theoretical basis, showing that the $\ell_2$ norm of learned features holds information useful for separating ID from OOD samples. NAN (Park et al., 2023) explained why this separation exists and introduced a negative-aware norm metric that captures both activation and deactivation tendencies. Block Selection (Yu et al., 2023) builds on Fang's criterion, selecting the most informative network blocks for $\ell_2$ norm scoring and demonstrating that certain blocks offer superior OOD separability.

**Based on Training Data Augmentation.** The methods in this category aim to enrich the training set. Outlier Exposure (OE) (Hendrycks et al., 2018) was a pioneering method, increasing robustness by incorporating auxiliary datasets into training. Later works improved this idea by mining hard negatives (Ming et al., 2022b; Chen et al., 2021) or generating near-OOD examples with generative models like GANs (Marek et al., 2021). While these techniques improve model calibration and resilience, they depend on the selection of suitable OOD samples, which may not represent the types of OOD data encountered during deployment. This makes them less feasible in settings where future OOD distribution is unknown.

**Based on Open-World Vision-Language Models.** The emergence of vision-language models (VLMs) such as CLIP (Radford et al., 2021) has recently advanced OOD detection. These approaches leverage a pretrained joint vision-language embedding space to enable a richer understanding of semantic concepts and support zero-shot OOD detection. Early works, such as ZOC (Esmaeilpour et al., 2022) and MCM (Ming et al., 2022a), demonstrate that CLIP can distinguish OOD samples by comparing image features with textual embeddings. Subsequent approaches (Wang et al., 2023; Nie et al., 2024) introduce learnable negative prompts and negation semantics to more

directly identify unknown samples. Despite their effectiveness, these VLM methods typically require significant computational resources for both fine-tuning and deployment. Additionally, many real-world applications do not demand full open-world recognition, making the overhead of such large-scale models unnecessary.

**Based on Input Perturbation.** Our work fits into this category, which amplifies the difference in model response between ID and OOD samples by applying perturbations. Earlier methods generally added small perturbations during inference. The intuition is that a model, being trained on ID data, will show limited output change when those inputs are perturbed, while its predictions for unfamiliar OOD samples are more sensitive to such changes. ODIN (Liang et al., 2017) is a well-known method in this direction, and G-ODIN (Hsu et al., 2020) extends it to a more general probabilistic framework. PRO (Chen et al., 2025) recently introduced an adversarial scoring function, noting that OOD confidence drops more sharply under perturbation.

Furthermore, several recent approaches also explore how to improve the OOD robustness of neural networks under adversarial perturbations. ATOM (Chen et al., 2021) jointly performs outlier mining and adversarial training, where worst-case perturbations of mined hard outliers from auxiliary data, thereby pushing these borderline samples away from the ID region and tightening the decision boundary. Similarly, RODEO (Mirzaei et al., 2025) generates near-distribution outliers via a diffusion model guided by CLIP similarity, and strengthens robustness by training the classifier with adversarial perturbations applied to these synthesized outliers so that the classifier learns to resist both natural and adversarial OOD variants. In contrast AROS (Mirzaei & Mathis, 2024) takes a different perspective by modeling embedding-space dynamics using Neural ODEs and employing stable synthetic embeddings to regularize the feature space under perturbations.

Notably, these methods generally rely on post-hoc techniques that require a separate perturbation to be calculated for each test input. In contrast, our approach trains a universal adversarial map that can be applied to all test samples, eliminating the need to generate new perturbations during inference. This universal map is added directly to the test data, resulting in a more efficient solution.

## 3 METHOD: ACTIVELY ENLARGING THE FEATURE NORM RATIO

Recent research has shown that neural networks tend to produce feature representations with significantly larger $l_2$ norms for ID data compared to OOD data (Park et al., 2023; Yu et al., 2023). Our work proposes a Universal Adversarial Training (UAT) framework that not only relies on this gap but actively and systematically amplifies it.

Let $f_\theta : X \to \mathbb{R}^d$ be the feature extractor of a model with parameters $\theta$. For a standard classifier, the logit $z_k$ for class $k$ is given by $z_k(x) = W_k^T f_\theta(x) + b_k$, where $W_k$ is the weight vector for class $k$. This can be expressed as:

$$z_k(x) = \|W_k\| \cdot \|f_\theta(x)\| \cdot \cos(\alpha_k(x)) + b_k \tag{1}$$

where $\alpha_k(x)$ is the angle between the feature vector $f_\theta(x)$ and the class prototype vector $W_k$. A high-confidence classification requires a large logit value, which depends on both a large feature norm $\|f_\theta(x)\|$ and a high degree of alignment ($\cos(\alpha_k) \to 1$).

The UAT objective is to solve the robust optimization problem:

$$\theta_{\text{uat}} = \arg\min_\theta \ \mathbb{E}_{(x,y)\sim P_{\text{in}}} \left[ \max_{L_{adv}} L_{\text{CE}}(z_\theta(x + L_{adv}), y) \right] \tag{2}$$

where $L_{adv}$ is a learnable universal adversarial map (UAM). Unlike traditional adversarial training, which computes a unique perturbation for each sample, UAT identifies and regularizes against a shared, systematic vulnerability in the model. This process acts as a powerful global regularizer, fundamentally reshaping the feature space.

Figure 1 illustrates the influence of UAT on the classifier's decision boundary during training. Initially, the boundary is highly irregular, and samples from different classes lie in close proximity. As the universal perturbation imposes repeated global pressure, the model gradually adapts its internal

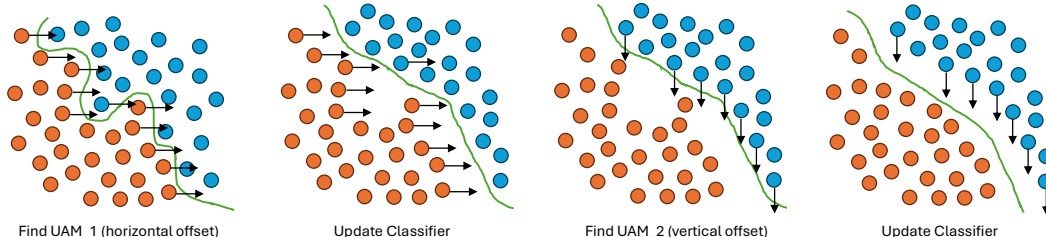

Find UAM_1 (horizontal offset)     Update Classifier     Find UAM_2 (vertical offset)     Update Classifier

Figure 1: The images from left to right illustrate how universal adversarial training progressively straightens the decision boundary, leading to more compact intra-class features and greater separation between classes. In the visualization, the black arrows indicate the universal perturbation. For clarity, we display the perturbations only for samples near the decision boundary so that the figure remains readable and free of visual clutter.

representations to resist this perturbation. As a result, samples within the same class become more compact, and the decision boundary smoothens, achieving clearer separation between classes.

## 3.1 THEORETICAL FOUNDATION

We begin by establishing a theoretical framework that explains why UAT leads to an enhanced feature norm-based separation for OOD detection. This reshaping can be characterized by two primary effects:

**Lemma 1** (Intra-Class Alignment and Compactness). *UAT induces a feature representation in which feature vectors of the same class become highly aligned with their class prototype vector, meaning the angle $\alpha_k(x)$ between a feature vector $f_\theta(x)$ and its corresponding prototype vector $W_k$ approaches zero, i.e., $\cos(\alpha_k(x)) \to 1$. A direct consequence of this universal alignment is that the feature vectors are mapped into a more compact region.*

**Lemma 2** (Inter-Class Separability). *UAT encourages the representations of different classes to be more separable in the feature space, increasing the distance between class-conditional means.*

We provide the detailed proofs of Lemma 1 and Lemma 2 in Appendix A.1 and Appendix A.2, respectively. Together, they illustrate that UAT reshapes the feature space into a highly structured representation where ID classes occupy compact, well-separated "islands". This structure is a direct consequence of enforcing robustness against a universal adversary. Building upon these geometric properties, we can now state our main proposition regarding the effect on feature norms for ID versus OOD samples.

**Proposition 1** (Feature Norm Ratio Amplification). *Let $\theta_{std}$ and $\theta_{uat}$ be the parameters of a model trained with standard cross-entropy and UAT, respectively. Then, the ratio of the expected feature norms between ID and OOD data is greater for the UAT-trained model:*

$$\frac{\mathbb{E}_{x \sim ID}[\|f_{\theta_{uat}}(x)\|]}{\mathbb{E}_{x' \sim OOD}[\|f_{\theta_{uat}}(x')\|]} > \frac{\mathbb{E}_{x \sim ID}[\|f_{\theta_{std}}(x)\|]}{\mathbb{E}_{x' \sim OOD}[\|f_{\theta_{std}}(x')\|]} \tag{3}$$

The detailed proof is provided in Appendix A.3.

## 3.2 TRAINING PROCEDURE

Our training process jointly optimizes the classifier $M$ and the universal adversarial map $L_{\text{adv}}$ in three phases, as outlined in Algorithm 1 (Appendix A.4). This procedure is the practical implementation of the robust optimization objective described in our theoretical framework.

**Phase 1**: Standard supervised learning is performed on clean input samples $\mathbf{x}$ with ground-truth labels $\mathbf{y}$. The model parameters $\theta$ are updated using the standard cross-entropy loss $\mathcal{L}_{\text{CE}}(M(\mathbf{x}), \mathbf{y}; \theta)$.

**Phase 2**: The learnable perturbation $L_{\text{adv}}$ is added to the input, resulting in $\mathbf{x} + L_{\text{adv}}$ during supervised learning. This phase encourages the model to remain invariant to structured, input-wide perturbations, promoting robustness in the feature space.

Figure 2: An overview of our channel-calibrated feature norm strategy to calculate the OOD score during inference.

**Phase 3**: The model parameters are frozen, and $L_{\text{adv}}$ is updated by applying the Projected Gradient Descent (PGD) adversarial attack (Madry et al., 2017). Gradients are computed to maximize the cross-entropy loss. This ensures that $L_{\text{adv}}$ continues to evolve as an effective perturbation throughout training, with its strength controlled by a multiplicative coefficient $\alpha$, while maintaining stable and synchronized training between the model and the adversarial map.

By interleaving these phases and controlling the frequency at which $L_{\text{adv}}$ is updated (e.g., every $\tau$ iterations), we ensure that the model is exposed to a stable adversarial signal for a sufficient duration. This allows the model to effectively learn from the perturbation and adapt to $\mathbf{x} + L_{\text{adv}}$ as a known challenge. In contrast, if $L_{\text{adv}}$ is updated too frequently, the perturbation changes rapidly before the model has an opportunity to adapt, preventing effective learning and resulting in unstable training and poor convergence, as demonstrated in our ablation studies.

### 3.3 CHANNEL-CALIBRATED FEATURE NORM

After training, we further enhance norm-based OOD detection at inference by introducing a channel-wise feature norm reweighting inspired by the Squeeze-and-Excitation mechanism in SENet (Hu et al., 2018). In SENet, it is shown that not all feature channels contribute equally to the final prediction, and the importance of each channel may vary across different images. Motivated by this, we propose a simple, non-parametric method that adaptively emphasizes channels with strong, discriminative activations while down-weighting channels with weaker responses.

Given a feature map $f \in \mathbb{R}^{C \times H \times W}$, we first apply the ReLU function to remove all negative activations, then compute the rescaled channel as:

$$f'_{c,h,w} = w_c \cdot \text{ReLU}(f_{c,h,w}),\tag{4}$$

where $w_c = \max_{h,w} \text{ReLU}(f_{c,h,w})$ is the maximum activation value of channel $c$ across all spatial positions. Afterward, we calculate the overall feature norm by averaging the $\ell_2$ norm of each channel's rescaled feature map:

$$\text{Norm}(f') = \frac{1}{C} \sum_{c=1}^{C} \sqrt{\sum_{w=1}^{W} \sum_{h=1}^{H} (f'_{c,h,w})^2},\tag{5}$$

This approach calibrates the influence of highly activated and semantically meaningful channels, which often correspond to foreground or class-relevant regions in ID samples. At the same time, it diminishes the contribution of less informative or background channels. Because OOD samples typically lack clear semantic structure, the activations across different channels tend to be more uniformly low. As a result, the norm score for OOD inputs is reduced, enhancing their separability from ID data during detection.

## 4 EXPERIMENTS

In this section, we assess the effectiveness of our proposed method by comparing it against several baseline approaches on standard OOD detection benchmarks. Most of the comparison results are taken from (Yu et al., 2023) while some results are sourced from (Regmi, 2025). Additionally, we evaluated recent methods by running the official implementations of (Zhang & Xiang, 2023; Song et al., 2024; Gao et al., 2025; Chen et al., 2025; Xu et al., 2023; Liu et al., 2023b) with their default parameters to ensure a fair and comprehensive comparison. Our method consists of two

main components: the Universal Adversarial Training (UAT) framework and the Channel-Calibrated Feature Norm (CCFN) scoring strategy. For clarity, we denote the model trained solely with our adversarial objective as Ours (UAT), while our complete approach, which integrates UAT with the CCFN scoring strategy, is referred to as Ours (UAT + CCFN).

## 4.1 IMPLEMENTATION DETAILS

**Datasets and Backbones.** We conduct experiments using CIFAR-10 and CIFAR-100 as ID datasets, employing ResNet-18 (He et al., 2016), VGG-11 (Simonyan & Zisserman, 2014), and WRN-28-10 (Zagoruyko & Komodakis, 2016) backbones. For large-scale evaluation, we use ImageNet as the ID dataset with a ResNet-50 backbone. Following standard practice, input images are resized to 32×32 for CIFAR and 224×224 for ImageNet.

**Training Hyperparameters.** For the CIFAR experiments, ResNet-18 is trained for 200 epochs, WRN-28-10 for 400 epochs, and VGG-11 for 100 epochs. The ResNet-50 on ImageNet is trained for 100 epochs. We determined these durations as further training did not yield noticeable improvements. Across all experiments, we use the SGD optimizer with a momentum of 0.9, a weight decay of 0.001, and a batch size of 128. The learning rate is set to 0.1 for all models except for VGG-11, which uses 0.05. For our UAT framework, the update interval $\tau$ is 50, and the adversarial map multiplicative coefficient $\alpha$ is 0.1.

## 4.2 EVALUATION PROTOCOL

To evaluate OOD detection performance, we adopt the protocol from FN (Yu et al., 2023). Specifically, the test set of the ID dataset is mixed with samples from an individual OOD dataset to form the evaluation set. Performance is measured using two standard metrics: the Area Under the Receiver Operating Characteristic Curve (AUROC) and the False Positive Rate at 95% True Positive Rate (FPR95). For all experiments, we report the mean of three independent runs in the main paper. The corresponding standard deviations are provided in the Appendix A.13.

| Method | SVHN | | Textures | | LSUN-C | | LSUN-R | | iSUN | | Places365 | | Average | |
|---|---|---|---|---|---|---|---|---|---|---|---|---|---|---|
| | FPR↓ | AUC↑ | FPR↓ | AUC↑ | FPR↓ | AUC↑ | FPR↓ | AUC↑ | FPR↓ | AUC↑ | FPR↓ | AUC↑ | FPR↓ | AUC↑ |
| MSP (Hendrycks & Gimpel, 2016) | 52.12 | 92.20 | 59.47 | 89.56 | 32.83 | 95.62 | 48.35 | 93.07 | 50.30 | 92.58 | 60.70 | 88.42 | 50.63 | 91.91 |
| EN (Liu et al., 2020) | 30.47 | 94.05 | 45.83 | 90.37 | 7.21 | 98.63 | 23.62 | 95.93 | 27.14 | 95.34 | 43.67 | 90.29 | 29.66 | 94.10 |
| EN+REACT (Sun et al., 2021) | 40.54 | 90.54 | 48.61 | 88.44 | 15.12 | 96.86 | 27.01 | 94.74 | 30.57 | 93.95 | 44.99 | 89.37 | 34.47 | 92.32 |
| EN+DICE (Sun & Li, 2022) | 25.95 | 94.66 | 47.22 | 89.82 | 3.83 | 99.26 | 27.70 | 95.01 | 31.07 | 94.42 | 49.28 | 88.08 | 30.84 | 93.54 |
| DML+ (Zhang & Xiang, 2023) | 33.16 | 90.59 | 27.01 | 92.86 | 5.98 | 98.71 | 48.72 | 88.89 | 23.41 | 94.89 | 47.01 | 87.78 | 30.88 | 92.45 |
| RFW (Song et al., 2024) | 54.81 | 87.67 | 48.81 | 87.50 | 15.89 | 96.75 | 36.58 | 92.59 | 36.31 | 92.70 | 47.19 | 87.75 | 39.93 | 90.83 |
| OTOD (Gao et al., 2025) | 22.87 | 91.25 | 41.59 | 88.92 | 5.89 | 97.48 | 30.72 | 94.64 | 32.74 | 94.30 | 41.34 | 89.51 | 29.19 | 92.68 |
| ODIN (Liang et al., 2017) | 33.83 | 93.03 | 45.49 | 90.01 | 7.29 | 98.62 | 20.05 | 96.56 | 23.09 | 96.01 | 45.06 | 89.86 | 29.14 | 94.02 |
| FN (Yu et al., 2023) | 7.13 | 98.65 | 31.18 | 92.31 | **0.07** | **99.96** | 27.08 | 95.25 | 26.02 | 95.38 | 62.54 | 84.62 | 25.67 | 94.36 |
| PRO (Chen et al., 2025) | 19.82 | 94.28 | 35.48 | 90.65 | 0.56 | 99.85 | 29.97 | 95.78 | 14.55 | 97.56 | **32.49** | 90.43 | 22.15 | 94.76 |
| Ours (UAT) | **6.98** | **98.84** | 15.40 | 97.28 | 0.15 | 99.95 | 10.80 | 97.88 | 6.42 | 98.23 | 49.32 | 91.25 | 14.85 | 97.24 |
| Ours (UAT + CCFN) | 7.25 | 98.72 | **15.13** | **97.34** | 0.37 | 99.91 | **6.51** | **98.77** | **5.09** | **99.05** | 38.92 | **93.04** | **12.21** | **97.80** |

Table 1: OOD detection results on ResNet-18 with CIFAR-10 as the ID dataset. The best and second-best results are highlighted in bold and underlined, respectively.

## 4.3 EVALUATION ON CIFAR-10 AND CIFAR-100

Following the setup of (Yu et al., 2023), we use CIFAR-10 and CIFAR-100 as ID datasets and evaluate OOD detection on six common benchmarks: SVHN (Netzer et al., 2011), Textures (Cimpoi et al., 2014), LSUN (crop) (Yu et al., 2015), LSUN (resize), iSUN (Xu et al., 2015), and Places365 (Zhou et al., 2017).

Table 1 shows that UAT consistently improves OOD detection. Compared to PRO (Chen et al., 2025), our UAT actively enlarges the feature norm gap to separate ID and OOD data. This improvement is significant, as the average FPR drops from 22.15% to 14.85%, and AUC increases from 94.76% to 97.24%. Adding CCFN further improves performance by suppressing less informative background features and emphasizing dominant channel activations, resulting in average 12.21%

FPR and 97.8% AUC. This advantage is particularly pronounced on datasets with complex backgrounds, such as iSUN, LSUN (resize), and Places365, where traditional norm-based methods are often misled by background activations. However, on LSUN (crop), the gain from CCFN is limited, as highlighted in Table 1. We attribute this to tightly cropped content in the dataset, which has largely reduced the presence of background pixels. Similarly, slight drops are seen on SVHN, likely due to its simple digit structure and low background complexity, which may limit the effectiveness of channel recalibration.

While Table 1 provides a detailed analysis under a simple setting, Table 2 presents a comparison of our method and baseline approaches across different architectures, including evaluations on CIFAR-100 dataset. Across VGG11, WRN-28-10, and ResNet18 backbones (for both CIFAR-10 and CIFAR-100), our method consistently outperforms existing baselines. For conciseness, we report average scores across OOD datasets in the main text, while detailed per-dataset results are provided in the Appendix A.13.

| | CIFAR-100 | | CIFAR-10 | | | |
| Method | ResNet18 | | WRN-28-10 | | VGG11 | |
| | FPR↓ | AUC↑ | FPR↓ | AUC↑ | FPR↓ | AUC↑ |
|---|---|---|---|---|---|---|
| MSP (Hendrycks & Gimpel, 2016) | 73.02 | 82.43 | 41.49 | 91.84 | 64.77 | 88.73 |
| EN (Liu et al., 2020) | 71.45 | 84.96 | 28.65 | 91.99 | 46.46 | _91.67_ |
| EN+REACT (Sun et al., 2021) | 70.95 | 84.79 | 86.22 | 65.78 | 47.15 | 88.44 |
| EN+DICE (Sun & Li, 2022) | 70.78 | 85.17 | 31.53 | 89.30 | 50.80 | 90.98 |
| DML+ (Zhang & Xiang, 2023) | 75.79 | 81.23 | 32.13 | 90.88 | 58.25 | 87.06 |
| RFW (Song et al., 2024) | 77.55 | 77.18 | 23.86 | 93.82 | 48.31 | 89.45 |
| OTOD (Gao et al., 2025) | 63.87 | _85.94_ | 29.22 | 90.95 | 47.52 | 91.56 |
| ODIN (Liang et al., 2017) | 60.44 | 78.69 | 27.17 | 94.93 | 48.21 | 87.08 |
| FN (Yu et al., 2023) | 60.27 | 84.09 | _13.53_ | _97.33_ | _39.34_ | 91.18 |
| PRO (Chen et al., 2025) | _60.03_ | 85.43 | 17.64 | 96.28 | 46.40 | 89.81 |
| Ours (UAT+CCFN) | **45.10** | **86.91** | **6.90** | **98.64** | **35.34** | **93.24** |

Table 2: Our method is evaluated on multiple network architectures using CIFAR-10 and CIFAR-100 as ID datasets. In the table, the best and second-best results are indicated in bold and underlined. For brevity, only average results are reported here; comprehensive results for each individual OOD dataset are available in the Appendix A.13.

### 4.4 Evaluation on ImageNet

Beyond CIFAR-10 and CIFAR-100, we further demonstrate the effectiveness of our approach on a more complex condition. With ImageNet as the ID dataset, we conduct experiments using ResNet-50 on four OOD benchmarks: iNaturalist (Van Horn et al., 2018), SUN (Xiao et al., 2010), Places365 (Zhou et al., 2017), and OpenImage-O (Wang et al., 2022). As shown in Table 3, our method considerably outperforms the second-best baseline, achieving an average improvement of 7.60% in FPR95 and 1.63% in AUROC.

## 5 Ablation Study

### 5.1 Impact of UAT on Feature Norm Ratios

We validate the proposed proposition (Eq. 3) through experiments by measuring feature norms and assessing OOD detection performance. Our evaluation setup is consistent with Table 1. To distinguish between different model and input settings, we denote feature extractors trained without universal adversarial perturbation as $f_{std}$, and those trained with it as $f_{uat}$. The input to $f$ can be either a clean test sample $\mathbf{x}$ or a perturbed sample $\mathbf{x} + L_{adv}$, leading to four experimental conditions in total. Table 4 presents the average feature norms for ID and OOD datasets, the ratio of ID to OOD norms, and the corresponding OOD detection performance. We find that the ID-to-OOD norm ratio serves as a strong indicator of OOD discriminability: when ID samples exhibit higher norms than OOD samples, the separation between the two becomes clearer, thus improving detection.

The results demonstrate that UAT substantially enlarges the norm ratio between ID and OOD data. Applying $L_{adv}$ during inference further increases this ratio. However, when $L_{adv}$ is applied to

| Method | iNaturalist | | SUN | | PLACES | | OpenImage-O | | Average | |
|---|---|---|---|---|---|---|---|---|---|---|
| | FPR↓ | AUC↑ | FPR↓ | AUC↑ | FPR↓ | AUC↑ | FPR↓ | AUC↑ | FPR↓ | AUC↑ |
| MSP (He et al., 2016) | 54.99 | 87.74 | 70.83 | 80.86 | 73.99 | 79.76 | 95.25 | 49.95 | 73.76 | 74.58 |
| EN (Liu et al., 2020) | 55.72 | 89.95 | 59.26 | 85.89 | 64.92 | 82.86 | 95.56 | 50.25 | 68.86 | 77.24 |
| EN+DICE (Sun & Li, 2022) | 25.63 | 94.49 | 35.15 | 90.83 | 46.49 | 87.48 | 90.52 | 55.37 | 49.45 | 83.04 |
| RFW (Song et al., 2024) | 69.51 | 85.84 | 73.50 | 80.71 | 76.62 | 78.54 | 88.02 | 59.47 | 76.91 | 76.14 |
| ODIN (Liang et al., 2017) | 47.66 | 89.66 | 60.15 | 84.59 | 67.89 | 81.78 | 94.89 | 49.29 | 67.65 | 76.33 |
| FN (Yu et al., 2023) | 22.01 | 95.76 | 42.93 | 90.21 | 56.80 | 84.99 | 88.13 | 58.11 | 52.47 | 82.27 |
| PRO (Chen et al., 2025) | 45.27 | 90.03 | 61.68 | 85.24 | 62.83 | 83.28 | 94.16 | 50.72 | 65.98 | 77.32 |
| NAC-UE (Liu et al., 2023b) | 20.39 | 95.57 | 37.04 | 89.44 | 51.75 | 85.08 | 90.29 | 58.46 | 49.87 | 82.14 |
| SCALE (Xu et al., 2023) | 39.13 | 86.34 | - | - | 76.25 | 70.04 | 60.66 | 81.10 | - | - |
| AdaSCALE-L (Regmi, 2025) | 32.72 | 88.82 | - | - | 73.90 | 70.37 | 53.84 | 83.09 | - | - |
| Ours(UAT) | 18.58 | 95.84 | 36.81 | 91.42 | 44.94 | 88.62 | 86.39 | 59.75 | 46.68 | 83.91 |
| Ours(UAT + CCFN) | 18.46 | 96.11 | 34.75 | 91.93 | 43.79 | 89.26 | 85.77 | 60.23 | 45.69 | 84.39 |

Table 3: Comparison of OOD detection performance on ImageNet. The best and second-best results are indicated in bold and underlined, respectively.

models trained without adversarial objectives (i.e., $f_{std}$), the OOD detection performance degrades, as the perturbation is specifically developed for adversarially trained models. In addition, across all conditions, a larger norm ratio consistently corresponds to better separation between ID and OOD samples. This phenomenon directly justifies our objective of enlarging the feature norm gap between ID and OOD distributions. More detailed, per-dataset results are provided in the Appendix A.13.

| Method | Performance | | ID Norm | OOD Norm | Norm Ratio |
|---|---|---|---|---|---|
| | FPR↓ | AUC↑ | | | |
| $f_{std}(x)$ | 28.31 | 94.00 | 0.454 | 0.293 | 1.612 |
| $f_{std}(x + L_{adv})$ | 31.30 | 93.14 | 0.428 | 0.291 | 1.518 |
| $f_{uat}(x)$ | 15.71 | 96.98 | 0.200 | 0.107 | 1.907 |
| $f_{uat}(x + L_{adv})$ | 14.85 | 97.24 | 0.200 | 0.104 | 1.950 |

Table 4: We analyze the feature norm values of ID and OOD samples produced by the networks. The networks are trained with the standard method and our UAT. We also consider the cases with and without adding the learned perturbation map.

## 5.2 IMPACT OF CLASSIFICATION PERFORMANCE

A major benefit of intrinsic model signal-based methods is that the classifier remains effective for both OOD detection and ID classification. To examine whether UAT influences classification (Lemmas 1 and 2), we compare standard training (i.e., $f_{std}(x)$) with the proposed UAT (i.e., $f_{adv}(x)$) across three architectures, and show the results in Table 5. While UAT causes a slight reduction in training accuracy, we consistently observe improved test accuracy, indicating that the model generalizes better and is more robust under adversarial training.

The effectiveness of UAT lies in its targeted regularization of decision boundaries. In standard models, these boundaries are often highly jagged and susceptible to small, adversarial shifts. UAT specifically strengthens the model against shared, structural vulnerabilities. This process smooths the decision boundaries, enlarges the gaps between different class distributions, and reduces the overlap between classes, as illustrated in Figure 3.

| Method | Backbone | Train Acc | Test Acc |
|---|---|---|---|
| $f_{std}(x)$ | ResNet18 | 99.9±0.0 | 94.0±0.3 |
| $f_{uat}(x)$ | | 97.3±0.2 | 95.0±0.2 |
| $f_{std}(x)$ | WRN-28-10 | 99.9±0.0 | 95.1±0.1 |
| $f_{uat}(x)$ | | 97.8±0.2 | 96.1±0.1 |
| $f_{std}(x)$ | VGG11 | 99.9±0.1 | 89.7±0.3 |
| $f_{uat}(x)$ | | 96.3±0.1 | 90.9±0.1 |

Table 5: Our UAT not only enhances OOD detection but also improves classification performance. We evaluated its effectiveness on several network backbones, including ResNet-18, WRN-28-10, and VGG-11.

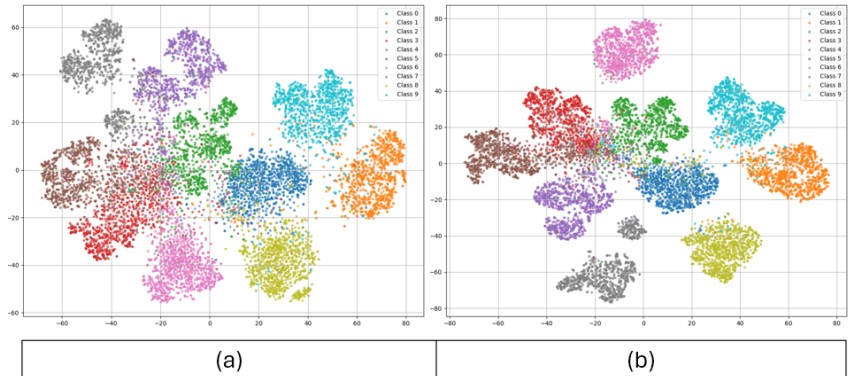

| (a) | (b) |

Figure 3: We employ t-SNE (Maaten & Hinton, 2008) to visualize the embeddings from the penultimate layer of a standard network (a) and a network trained with our UAT method (b). The visualization results demonstrate that UAT leads to more compact class-wise distributions and greater inter-class separability.

## 5.3 IMPACT OF UPDATING FREQUENCY ON UAM

We analyze how the update frequency $\tau$ of $L_{\text{adv}}$ during training impacts overall performance and summarize the results in Table 6. Notably, a smaller $\tau$ (i.e., more frequent updates) gives the model less time to adapt to a given adversarial signal, often resulting in unstable training and suboptimal OOD detection. Increasing $\tau$ allows the model to better learn from the perturbation and develop more discriminative features. However, setting $\tau$ too large can render the adversarial map outdated, making it less effective as the model parameters evolve. We also observe a strong correlation between OOD detection performance and the model's generalization, as measured by ID test classification accuracy. This suggests that conventional validation-based hyperparameter selection can be used to identify an appropriate setting for $\tau$.

|  | Performance | | Train | Test |
|---|---|---|---|---|
| $L_{\text{adv}}$ update frequency | FPR↓ | AUC↑ | Acc | Acc |
| $\tau = 3$ | 88.18±2.8 | 68.99±1.3 | 69.7±0.6 | 52.3±1.9 |
| $\tau = 10$ | 53.94±1.2 | 89.27±0.5 | 90.9±0.1 | 90.6±0.1 |
| $\tau = 25$ | 17.47±0.8 | 95.70±0.7 | 96.3±0.1 | 94.5±0.1 |
| $\tau = 50$ | 12.21±1.3 | 97.80±0.2 | 97.3±0.2 | 95.0±0.2 |
| $\tau = 100$ | 19.00±0.0 | 96.63±0.1 | 98.7±0.3 | 94.4±0.1 |

Table 6: We investigate the effect of update frequency ($\tau$) on adversarial maps during training, analyzing its impact on both classification and OOD detection performance.

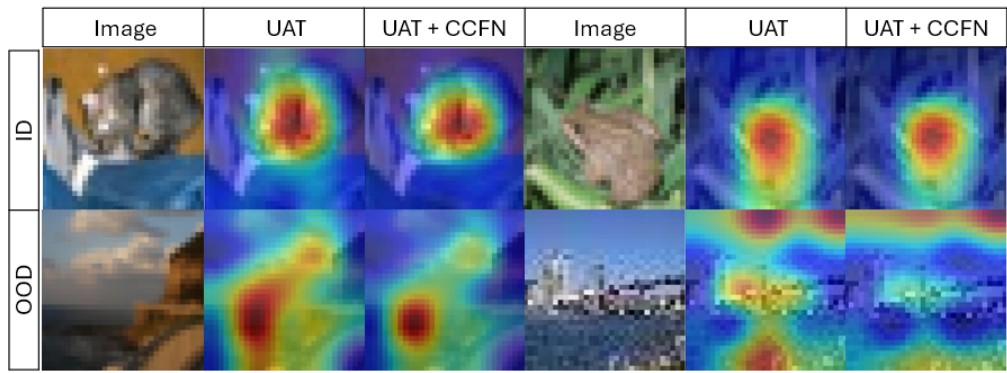

Figure 4: Visualization of the activation maps demonstrates that our channel calibration framework (CCFN) can effectively suppress background and irrelevant signals.

## 5.4 VISUALIZATION OF CHANNEL-CALIBRATED FEATURE MAPS

To further illustrate the effectiveness of our channel-calibrated feature norm strategy, we visualize feature activation maps with and without channel calibration. Figure 4 displays comparisons for both ID and OOD samples, where UAT and UAT + CCFN represent activation maps generated without and with channel calibration, respectively. In these heatmaps, blue denotes low activations and red denotes high activations. The CCFN method yields more concentrated activations on semantically meaningful objects for ID samples, whereas for OOD samples, CCFN effectively suppresses irrelevant signals, resulting in lower feature norms and improved OOD discrimination.

## 5.5 EVALUATION OF REAL-WORLD DATASETS.

To further assess the real-world applicability of our method, we evaluate it on two challenging distribution-shifted datasets: MIDOG (Aubreville et al., 2023) and SpaceNet-8 (Hänsch et al., 2022). MIDOG is a medical histopathology dataset, whereas SpaceNet-8 is a remote-sensing dataset, both containing domain-shifted OOD images. In this experiment, we also extend the proposed UAT framework to a ViT (Dosovitskiy, 2020) backbone. As shown in Table 7, UAT remains effective across both CNN and non-CNN architectures and demonstrates strong generalization to real-world distribution shifts that extend well beyond conventional OOD detection benchmarks. We report only the averaged performance across the evaluated OOD subsets, while detailed implementation settings and additional discussion for both datasets are provided in Appendix A.12.

| Method | MIDOG | | SpaceNet-8 | |
|---|---|---|---|---|
| | Avg. FPR↓ | Avg. AUC↑ | FPR↓ | AUC↑ |
| SCALE (RNet50) (Xu et al., 2023) | 58.48 | 76.44 | 25.29 | 86.68 |
| OTOD (RNet50) (Gao et al., 2025) | 56.51 | 73.90 | 30.57 | 83.81 |
| NAC-UE (RNet50) (Liu et al., 2023b) | 54.84 | 77.26 | 25.74 | 86.25 |
| CATEX (CLIP) (Liu et al., 2023a) | 64.57 | 75.90 | 29.33 | 82.19 |
| ViM (ViT)(Wang et al., 2022) | 51.01 | 78.17 | _23.17_ | _90.18_ |
| Ours (RNet50) | _48.96_ | _80.56_ | 24.63 | 88.46 |
| Ours (ViT) | **47.18** | **83.89** | **21.8** | **92.51** |

Table 7: OOD detection results on the MIDOG and SpaceNet-8 datasets. For MIDOG, we report the average performance across all OOD subsets. For SpaceNet-8, the non-flooded images are treated as ID, while the corresponding flooded images serve as OOD. The best and second-best results are highlighted in bold and underlined, respectively

## 5.6 CONCLUSIONS

In this paper, we introduced a novel framework for OOD detection that actively enlarges the feature norm gap between ID and OOD samples. Through theoretical analysis and extensive experiments, we demonstrated that our method achieves state-of-the-art performance. Our approach is centered around universal adversarial training, which learns a single, universal perturbation map to regularize the model's decision boundaries. This process enhances the model's generalization and makes it robust at distinguishing ID and OOD samples under both clean and perturbed conditions. To complement our training strategy, we proposed the channel-calibrated feature norm, a simple yet effective scoring function that refines feature representations at inference by emphasizing semantically important channels and suppressing background noise.

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

# A    APPENDIX

## A.1    PROOF OF LEMMA 1: INTRA-CLASS ALIGNMENT AND COMPACTNESS

**Lemma 1.** *UAT induces a feature representation in which feature vectors of the same class become highly aligned with their class prototype vector, meaning the angle $\alpha_k(x)$ between a feature vector $f_\theta(x)$ and its corresponding prototype vector $W_k$ approaches zero, i.e., $\cos(\alpha_k(x)) \to 1$. A direct consequence of this universal alignment is that the feature vectors are mapped into a more compact region.*

*Proof.* The UAT objective in Eq. 2 forces the model to be robust against a universal perturbation $L_{adv}$. Geometrically, this robustness is achieved by smoothing the decision boundary and pushing it away from the data manifold. This process incentivizes the class-conditional feature distribution to become more convex, where the classification outcome is insensitive to small perturbations. A distribution with a complex, non-convex shape would present numerous vulnerable points along its boundary, susceptible to a universal adversary from various directions. By forcing all intra-class variations into a single, simply-connected region, UAT leads to high alignment with the class prototype (as all features are now in a narrow cone around the prototype $W_k$), causing the alignment term $cos(\alpha_k)$ to approach its maximum of 1 and consequently reducing intra-class variance.

## A.2    PROOF OF LEMMA 2: INTER-CLASS SEPARABILITY

**Lemma 2.** *UAT encourages the representations of different classes to be more separable in the feature space, increasing the distance between class-conditional means.*

*Proof.* Robustness in UAT demands that the classification margin, $z_k(x + \delta) - z_j(x + \delta)$, remains positive and large for all $j \neq k$. To analyze this, we first approximate the margin.

**Step A. Margin Approximation.**    We first approximate the classification margin $z_k - z_j$ for a sample $x$ belonging to class $k$ and $j$. This appendix provides a detailed, step-by-step derivation of this approximation.

The classification margin is defined as:

$$z_k(x) - z_j(x) = (W_k^\top f_\theta(x) + b_k) - (W_j^\top f_\theta(x) + b_j)$$
$$z_k - z_j = (W_k - W_j)^\top f_\theta(x) + (b_k - b_j). \tag{6}$$

From the conclusion of Lemma 1 (Intra-Class Compactness), we established that a UAT-trained model learns to map features of a given class to a compact region, highly aligned with the corresponding class prototype vector $W_k$. We can formalize this high degree of alignment as:

$$f_\theta(x) \approx \|f_\theta(x)\| \frac{W_k}{\|W_k\|}. \tag{7}$$

This approximation states that the feature vector $f_\theta(x)$ has a direction nearly identical to that of its class prototype $W_k$.

Substituting equation 7 into equation 6:

$$z_k - z_j \approx (W_k - W_j)^\top \left( \|f_\theta(x)\| \frac{W_k}{\|W_k\|} \right) + (b_k - b_j). \tag{8}$$

We can expand the dot product term:

$$z_k - z_j \approx \|f_\theta(x)\| \left( \frac{W_k^\top W_k}{\|W_k\|} - \frac{W_j^\top W_k}{\|W_k\|} \right) + (b_k - b_j). \tag{9}$$

Using the definitions of the L2-norm ($\|A\|^2 = A^\top A$) and the dot product ($A^\top B = \|A\| \cdot \|B\| \cos \beta$), we can rewrite the terms:

$$z_k - z_j \approx \|f_\theta(x)\| \left( \|W_k\| - \|W_j\| \cos(\beta_{kj}) \right) + (b_k - b_j), \tag{10}$$

where $\beta_{kj}$ is the angle between the prototype vectors $W_k$ and $W_j$.

To simplify this expression further and isolate the geometric term, we introduce an assumption regarding the norms of the class prototype vectors. This assumption is well-grounded in empirical observations of deep networks. During the terminal phase of training, classifiers often exhibit a phenomenon known as *Neural Collapse*, where the norms of the prototype vectors for all classes converge to the same value (Papyan et al., 2020). Furthermore, standard practices like L2 weight regularization inherently penalize disproportionately large weight vectors, further encouraging this balancing. We thus assume $\|W_k\| \approx \|W_j\|$ and factor out $\|W_k\|$ from the expression:

$$z_k - z_j \approx \|f_\theta(x)\| \cdot \|W_k\|(1 - \cos(\beta_{kj})) + (b_k - b_j). \tag{11}$$

In modern deep networks, the separation between classes is primarily driven by the interaction between features and weights. The core task of a powerful model is to learn meaningful features $f_\theta(x)$ from the input $x$ and use these features for classification. If the model were to primarily rely on a large, input-independent bias term, $\Delta b = b_k - b_j$, to distinguish between classes, it would essentially mean the model has failed to learn meaningful representations. Therefore, for analyzing the geometric separation, we focus on the dominant, feature-dependent term:

$$z_k - z_j \approx \|f_\theta(x)\| \cdot \|W_k\|(1 - \cos(\beta_{kj})). \tag{12}$$

The above equation demonstrates that the classification margin is directly proportional to the feature norm, the prototype norm, and a geometric term $(1 - \cos(\beta_{kj}))$ showing the angular separation of the class prototypes. This justifies our argument that to create a robust margin under a regularized norm, the model must maximize this angular separation.

**Step B. Maximizing the Margin.** As established by the optimizer's implicit bias, UAT acts as a strong regularizer that constrains the feature norm $\|f_\theta(x)\|$ from growing uncontrollably (Roth et al., 2020). With the feature norm being regularized, the model must find an alternative way to create a robust margin. From the approximation in Eq. 12, the most effective way to robustly maximize the margin is to maximize the geometric term $(1 - \cos(\beta_{kj}))$. This is achieved by maximizing the angle $\beta_{kj}$ between the prototype vectors. Since the class-conditional mean $\mu_k$ is directionally aligned with its prototype $W_k$ (from Lemma 1), separating the prototypes directly leads to an increased Euclidean distance $\|\mu_k - \mu_j\|^2$ between the class means.

## A.3 Proof of Proposition

**Proposition 1** (Feature Norm Ratio Amplification). *Let $\theta_{std}$ and $\theta_{uat}$ be the parameters of a model trained with standard cross-entropy and UAT, respectively. Then, the ratio of the expected feature norms between ID and OOD data is greater for the UAT-trained model:*

$$\frac{\mathbb{E}_{x \sim ID}[\|f_{\theta_{uat}}(x)\|]}{\mathbb{E}_{x' \sim OOD}[\|f_{\theta_{uat}}(x')\|]} > \frac{\mathbb{E}_{x \sim ID}[\|f_{\theta_{std}}(x)\|]}{\mathbb{E}_{x' \sim OOD}[\|f_{\theta_{std}}(x')\|]} \tag{13}$$

*Proof.* The UAT process exerts asymmetric pressures on the feature norms of ID and OOD samples.

**Pressure on ID Norms:** The ID norm $\|f_{\text{ID, uat}}\|$ is subject to a tug-of-war. UAT, as a strong regularizer, applies a general downward pressure ($\gamma_{\text{reg}} < 1$) on all feature norms(Grathwohl et al., 2019). However, to satisfy the classification objective, the model must maintain a sufficient logit value. As established by Lemma 1, the alignment $\cos(\alpha_k)$ for ID samples is pushed to its maximum. Therefore, the feature norm $\|f_{\text{ID}}\|$ becomes the primary lever to ensure the classification margin, creating an upward "support" pressure ($\gamma_{\text{ce}} > 1$) that prevents the norm from collapsing.

**Pressure on OOD Norms:** The OOD norm $\|f_{\text{OOD, uat}}\|$ is subject to the same general downward regularization pressure as ID samples. However, a critical difference arises: OOD samples lack the "upward support" pressure that benefits ID samples. Since OOD data is not part of the training objective, there is no classification task that requires the model to maintain a large feature norm for these inputs. This asymmetry is a direct consequence of the model learning a sharpened definition of the ID manifolds, as established by Lemmas 1 and 2.

To model this asymmetric effect, both norms are suppressed by a general factor $\gamma_{\text{reg}}$, while the ID norm experiences a less effective suppression due to the presence of a classification objective. The ratio for the UAT model is therefore:

$$\frac{\mathbb{E}_{x \sim ID}[\|f_{\theta_{\text{uat}}}(x)\|]}{\mathbb{E}_{x' \sim OOD}[\|f_{\theta_{\text{uat}}}(x')\|]} \approx \frac{\gamma_{ce} \cdot \gamma_{reg} \cdot \mathbb{E}_{x \sim ID}[\|f_{\theta_{\text{std}}}(x)\|]}{\gamma_{reg} \cdot \mathbb{E}_{x' \sim OOD}[\|f_{\theta_{\text{std}}}(x')\|]} > \frac{\mathbb{E}_{x \sim ID}[\|f_{\theta_{\text{std}}}(x)\|]}{\mathbb{E}_{x' \sim OOD}[\|f_{\theta_{\text{std}}}(x')\|]} \quad (14)$$

### A.4 PSEUDO CODE

Algorithm 1 presents the detailed pseudocode of the proposed multi-phase adversarial training procedure, which underpins the methodology described in Section 3.2

---

**Algorithm 1** Adversarial Training with Multi-Phase Perturbation Strategy

---

**Require:** Training dataset $\mathcal{D} = (X, Y)$, model $M$, adversarial map $L_{\text{adv}}$, optimizer, loss function $\mathcal{L}_{\text{CE}}$, Adversarial update interval $\tau$, Adversarial attack $\gamma$, coefficient $\alpha$

1: **for** each epoch $t = 1$ to $T$ **do**
2:     Shuffle the training dataset $\mathcal{D}$ into $(\mathcal{B}_1, \ldots, \mathcal{B}_T)$
3:     **for** each batch $\mathcal{B}_t \in \{\mathcal{B}_1, \ldots, \mathcal{B}_T\}$ **do**
4:         **if** $\mathcal{B}_t \% 2 == 0$ **then**
            *Phase 1: Clean training*
5:             $y_{\text{pred}} \leftarrow M(x)$ for $(x, y) \in \mathcal{B}_t$
6:             Update $M$ by minimizing $\mathcal{L}_{\text{CE}}(y_{\text{pred}}, y)$
7:         **else if** $\mathcal{B}_t \% 2 == 1$ **then**
            *Phase 2: Training under perturbation*
8:             $y_{\text{pred}} \leftarrow M(x + L_{\text{adv}})$ for $(x, y) \in \mathcal{B}_t$
9:             Update $M$ by minimizing $\mathcal{L}_{\text{CE}}(y_{\text{pred}}, y)$
10:        **end if**
11:       **if** $\mathcal{B}_t \% \tau == 0$ **then**
            *Phase 3: Update adversarial map*
12:             Adversarial input: $x_{\text{adv}} \leftarrow \gamma(M(x))$
13:             $y_{\text{pred}} \leftarrow M(x_{\text{adv}} + L_{\text{adv}})$ for $(x, y) \in \mathcal{B}_t$
14:             Update $\alpha L_{\text{adv}}$ by maximizing $\mathcal{L}_{\text{CE}}(y_{\text{pred}}, y)$
15:        **end if**
16:     **end for**
17: **end for**

---

### A.5 IMPACT OF UAM POSITIONS

We investigate the impact of applying UAM at various network depths, specifically at the input, middle, and final layers. As shown in Table 8, introducing the perturbation at the input layer yields the most significant improvement. One possible explanation is that perturbations injected at the input layer are propagated through the entire network, resulting in a cascading and amplified effect. This forces the model to learn robust features from the ground up. Conversely, a perturbation applied to deeper layers directly attacks the already-formed abstract features near the decision boundary. At

this late stage, the network lacks the subsequent processing capacity to recover or build a new robust representation. Consequently, this leads to a weaker training signal for effective OOD separation.

| Method | Performance | |
|--------|-------------|---|
| | FPR↓ | AUC↑ |
| Front | 12.21±1.3 | 97.80±0.2 |
| Middle | 18.59±0.6 | 96.43±0.2 |
| Final | 22.16±0.4 | 96.17±0.1 |

Table 8: We analyze the performance of our UAT when the perturbation map is applied at different network positions.

## A.6 PER-SAMPLE VS. UNIVERSAL PERTURBATION STRATEGIES

We compare two adversarial training strategies: (1) using a universal perturbation optimized across the entire dataset, and (2) applying individual perturbations to each sample. Both strategies were implemented using projected gradient descent (PGD) attacks (Madry et al., 2017). As shown in Table 9, training with a universal perturbation yields notably stronger OOD detection than per-sample PGD. The universal map is able to capture shared vulnerabilities and accentuate structural inconsistencies that are common to OOD data, while per-sample PGD tends to focus on individual instances, missing broader and more generalizable cues. Moreover, the universal perturbation strategy is more computationally efficient, as it requires optimization only once per dataset rather than once per sample.

| Method | Performance | |
|--------|-------------|---|
| | FPR↓ | AUC↑ |
| Per-sample Adversarial | 22.91±0.8 | 95.83±0.2 |
| Universal Advesarial | 12.21±1.3 | 97.80±0.2 |

Table 9: A performance comparison between per-sample adversarial training and our universal adversarial training, with both types of adversaries generated using the PGD attack.

## A.7 NEAR OOD DETECTION

To compare with prior works, we adopt an established protocol for far-distribution OOD detection in the main paper. However, a more challenging and practical scenario involves near-distribution OOD detection, where the semantic gap between ID and OOD data is small. This experiment was conducted to evaluate the robustness and effectiveness of our method under these difficult conditions.

The results in Table 10 show a performance comparison of various methods using CIFAR-10 as the ID dataset and CIFAR-100 as the near-OOD dataset. Our proposed approach, combining UAT and CAFN, achieves the highest performance across both FPR95 and AUROC metrics, demonstrating its superior ability to distinguish between fine-grained semantic differences.

To further validate our approach beyond the standard near-OOD setup, we also consider the recently introduced adjacent OOD detection benchmark by (Yang et al., 2025), which spans three datasets: Faces (Erhan et al., 2013), Cars (Krause et al., 2013), and Food (Bossard et al., 2014). Unlike conventional OOD benchmarks, this setting is particularly difficult as the ID and OOD distributions exhibit substantial overlap. As shown in Table 10, our method achieves higher ID generalization across these datasets compared to the baseline, while also delivering superior OOD detection performance.

## A.8 IMPACT OF PERTURBATION STRENGTH COEFFICIENT($\alpha$)

The perturbation strength coefficient, $\alpha$, is a important hyperparameter that balances the trade-off between model robustness and feature discriminability. This analysis was performed to investigate

| Method | Performance | |
|---|---|---|
| | FPR↓ | AUC↑ |
| MSP (He et al., 2016) | 54.83 | 85.49 |
| EN (Liu et al., 2020) | 43.58 | 89.18 |
| EN+DICE (Sun & Li, 2022) | 51.71 | 86.53 |
| RFW (Song et al., 2024) | 45.94 | 90.22 |
| ODIN (Liang et al., 2017) | 88.27 | 72.06 |
| FN (Yu et al., 2023) | 48.94 | 87.62 |
| DHM (Cao & Zhang, 2022) | 47.28 | 88.51 |
| Ours (UAT + CCFN) | **42.02** ± 0.2 | **92.17** ± 0.1 |

Table 10: Comparison of Near OOD performance with CIFAR-10 as ID vs CIFAR-100 as OOD. The best and second-best results are indicated in bold and underlined, respectively.

| Method | Faces | | | Cars | | | Food | | |
|---|---|---|---|---|---|---|---|---|---|
| | FPR↓ | AUC↑ | Test Acc | FPR↓ | AUC↑ | Test Acc | FPR↓ | AUC↑ | Test Acc |
| Supervised(MSP) | 88.2 | 70.8 | 74% | 88.8 | 69.2 | 80% | 81.1 | 78.8 | 74% |
| SimCLR KNN | 95.0 | 52.0 | | 94.0 | 52.5 | | 91.6 | 61.1 | |
| SimCLR SSD | 95.1 | 55.0 | | 93.1 | 52.7 | | 89.3 | 64.4 | |
| RotLoss KNN | 95.8 | 46.1 | | 94.8 | 51.1 | | 94.0 | 49.7 | |
| RotLoss SSD | 95.7 | 46.6 | | 95.0 | 50.7 | | 94.9 | 50.7 | |
| UAT(MSP) | 87.1 | 70.8 | 76% | 88.2 | 70.4 | 82% | 81.1 | 79.5 | 75% |
| UAT(FN) | **86.7** | **72.2** | 76% | **86.0** | **71.3** | 82% | **80.7** | **81.3** | 75% |

Table 11: OOD detection performance and test accuracy reproduced from (Yang et al., 2025) on the adjacent OOD benchmark. We additionally report test accuracy to show that our method to show improved ID generalization for near OOD datasets. Baseline results are also taken from (Yang et al., 2025).

the sensitivity of our model to this parameter and to identify an optimal operating range, ensuring that our reported performance is not contingent on a single, fine-tuned value.

Table 2 investigates the influence of $\alpha$ on OOD detection performance. The results indicate that a moderate perturbation strength ($\alpha = 0.1$) yields an optimal balance, achieving the best performance. Excessively strong perturbations can disrupt the learning of core ID features, while overly weak perturbations may not provide a sufficient regularization signal. This finding highlights the importance of a well-calibrated perturbation for effective training.

| Perturbation strength coefficient $\alpha$ | Performance | |
|---|---|---|
| | FPR↓ | AUC↑ |
| $\alpha = 1$ | 15.34 ± 0.8 | 96.09 ± 0.2 |
| $\alpha = 0.5$ | 14.42 ± 0.9 | 96.71 ± 0.5 |
| $\alpha = 0.1$ | 12.21 ± 1.3 | 97.80 ± 0.2 |
| $\alpha = 0.01$ | 17.52 ± 1.8 | 95.47 ± 0.6 |

Table 12: We investigate the effect of perturbation strength $\alpha$ on model's OOD detection performance.

## A.9 VISUALIZATION OF INPUT PERTURBATION $L_{\text{ADV}}$

The Universal Adversarial Map (UAM) is the core component of our Universal Adversarial Training (UAT) framework. We were interested in visualizing the learned map, $L_{\text{adv}}$, to gain qualitative insight into the nature of the perturbation. As Figure 5 illustrates, however, the visualized UAM appears as a subtle, high-frequency noise map without obvious, human-interpretable structures. While

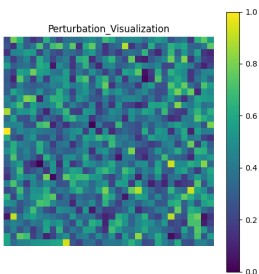

Figure 5: Visualization of the learned universal adversarial map $L_{\mathrm{adv}}$.

it does not reveal simple patterns, its effectiveness as a global regularizer suggests it exploits complex, high-dimensional vulnerabilities of the model. Understanding the underlying properties of such effective, yet seemingly unstructured, adversarial perturbations remains an interesting direction for future research.

## A.10  IMPACT OF CALIBRATION STRATEGIES

| Method | Performance | |
|---|---|---|
| | FPR↓ | AUC↑ |
| No calibration | 14.85 | 97.24 |
| L2 Norm calibration | 14.63 | 97.35 |
| Variance calibration | 14.97 | 97.31 |
| Mean calibration | 22.26 | 95.76 |
| Min channel calibration | 99.96 | 22.26 |
| Max channel calibration | **12.21** | **97.80** |

Table 13: We evaluate the performance various calibration strategies. The best results are highlighted in bold

In the Table 13, we analyze the effect of various calibration strategies on OOD performance. Among these strategies, max-channel calibration is particularly effective because modern deep feature maps tend to contain a small subset of channels that exhibit strong, semantically meaningful activations (Hu et al., 2018). Using the maximum activation as the rescaling factor amplifies these highly discriminative channels while naturally suppressing weak or noisy responses that often correspond to background content. Regarding the aggregation method, we average the squared norms of the rescaled channels because this quantity has been shown to approximate a confidence measure for the hidden classifier, as demonstrated in (Park et al., 2023). This aggregation therefore provides a principled way to separate ID and OOD samples.

To further support this choice, we evaluated several alternative calibration strategies in this rebuttal, including mean activation, minimum activation, variance, and the L2 norm of the feature map. The results in Table 13 show that the maximum-activation rescaling yields the strongest OOD detection performance. These empirical findings reinforce that the proposed max-channel calibration is not only intuitively aligned with the nature of deep feature representations but also empirically the most effective among the tested alternatives.

## A.11  TRAINING AND INFERENCE DETAILS.

Table 14 summarizes the training duration and average inference latency for standard training and UAM on ResNet-50 using ImageNet.The results show that our method introduces only a small overhead compared to standard approach. All experiments were conducted on a server equipped with 4× NVIDIA GeForce RTX 4070 Ti SUPER GPUs.

## A.12  DETAILED EVALUATION OF REAL-WORLD DATASETS.

| Method | Training Time (100 epochs) | Avg. Inference Latency (s/image) |
|---|---|---|
| Standard Training (L2 norm) | 25h 40m | 0.0123 |
| UAT (CCFN) | 27h 10m | 0.0127 |

Table 14: Training time and average inference latency per image for ResNet-50 on ImageNet.

In the Table 7,we only provided limited implementation details and results discussion regarding MIDOG (Aubreville et al., 2023) and SpaceNet-8 (Hänsch et al., 2022). Here we provide complete information for the Section 5.5

- The MIDOG (Mitosis Domain Generalization) dataset is a digital pathology dataset, focusing on mitotic figure detection in H&E-stained breast cancer tissue. In MIDOG, each sub-dataset represents a distinct domain with variations in imaging conditions. The domains differ in scanner type, staining protocol, image resolution, acquisition center, and overall visual style. Specifically, datasets 1a-c are sourced from different scanners within the same institution, datasets 2-7 introduce differences in staining protocols, imaging devices, acquisition centers, tissue preparation, and different cancer types from human and canine species. In our evaluation, we used 1a-c as our ID, whereas the rest is used as OOD.

- SpaceNet-8 is a large-scale remote sensing dataset for flood detection, containing multi-spectral satellite imagery from several geographic regions. In our experiment on SpaceNet-8, we treat non-flooded images from two cities as ID, and use flooded images as OOD samples to evaluate the model's ability to detect distributional shift.

We extended the UAT framework to the Vision Transformer (ViT) backbone to demonstrate the generality of our approach. The core logic of UAT remains applicable, but we adapt the OOD scoring mechanism to the ViT's feature space, following established practices for OOD detection in ViT. Specifically, after training ViT with UAT, we extract the penultimate-layer features and apply a channel-wise gating mechanism (max activation) to emphasize the most informative dimensions for inference. To calculate the norm values, we follow the ViM (Wang et al., 2022) formulation. We compute the feature center using the pseudo-inverse of the classifier weights and estimate the covariance of centered features. The smallest-variance eigenvectors form an orthogonal subspace, and projecting features onto this subspace yields the orthogonal projection norm, which quantifies deviation from the ID feature manifold.

**Experiment Setup.** To ensure a fair comparison, all baseline methods were run using their official repositories. Among these baselines, CATEX is a CLIP-based approach pretrained on ImageNet. To adapt CATEX to MIDOG and SpaceNet-8, we followed its original design and constructed domain- and label-specific textual prompts, then finetuned the model to identify visual patterns that deviate from the expected distribution, such as unfamiliar cellular textures or image domains unrelated to the training data.

**Results.** As shown in Tables 15 and 16, our method consistently yields substantial improvements over all competing approaches on both datasets. Notably, CATEX performs significantly worse than the other baselines, which we believe is due to the mismatch between the general-purpose semantics encoded in the ImageNet-pretrained CLIP backbone and the highly specialized, fine-grained visual cues required for medical pathology (MIDOG) and satellite imagery (SpaceNet-8). These results demonstrate that UAT remains effective across both CNN and non-CNN architectures and that it generalizes well to real-world distribution shifts that go far beyond conventional OOD detection benchmarks.

## A.13 DETAILED EXPERIMENTAL RESULTS

Due to space limitations, we reported only the averaged AUROC, FPR95 and norm ratio values of several experiments in the main paper. In this section, we provide a complete breakdown of results across all OOD datasets in Tables 17, 18, 19, 20, 21, 22, and Table 23, allowing for a more comprehensive comparison.

| Method | 2 | | 3 | | 4 | | 5 | | 6a | | 6b | | 7 | | Avg | |
|---|---|---|---|---|---|---|---|---|---|---|---|---|---|---|---|---|
| | FPR↓ | AUC↑ | FPR↓ | AUC↑ | FPR↓ | AUC↑ | FPR↓ | AUC↑ | FPR↓ | AUC↑ | FPR↓ | AUC↑ | FPR↓ | AUC↑ | FPR↓ | AUC↑ |
| SCALE (RNet50) (Xu et al., 2023) | 67.60 | 72.53 | 67.51 | 72.43 | 28.20 | 85.78 | 54.84 | 78.85 | 66.09 | 69.06 | 69.88 | 78.29 | 55.24 | 78.15 | 58.48 | 76.44 |
| OTOD (RNet50) (Gao et al., 2025) | 64.61 | 69.28 | 66.16 | 67.07 | 22.59 | 92.65 | 63.31 | 66.38 | 59.38 | 69.77 | 58.38 | 80.70 | 61.17 | 71.46 | 56.51 | 73.90 |
| NAC-UE (RNet50) (Liu et al., 2023b) | 59.08 | 75.91 | 59.61 | 77.80 | 30.14 | 82.46 | 56.23 | 71.79 | 57.99 | 76.61 | 61.88 | 80.96 | 58.95 | 75.28 | 54.84 | 77.26 |
| CATEX (CLIP) (Liu et al., 2023a) | 75.41 | 70.87 | 71.22 | 74.65 | 20.30 | 92.23 | 70.60 | 68.60 | 60.01 | 75.66 | 86.18 | 73.74 | 68.29 | 75.53 | 64.57 | 75.90 |
| ViM (ViT) (Wang et al., 2022) | 55.37 | 77.14 | 48.39 | 75.56 | 23.73 | 92.12 | 48.24 | 79.59 | 41.04 | 83.39 | 89.42 | 63.14 | 50.90 | 76.25 | 51.01 | 78.17 |
| Ours (RNet50) | 58.86 | 76.57 | 52.16 | 81.49 | 18.91 | 93.42 | 48.52 | 81.02 | 48.32 | 79.49 | 67.91 | 73.38 | 48.03 | 85.74 | 48.96 | 80.56 |
| Ours (ViT) | 51.20 | 81.29 | 53.17 | 81.91 | 17.91 | 94.95 | 49.05 | 82.21 | 44.93 | 85.80 | 60.14 | 80.54 | 53.89 | 80.50 | 47.18 | 83.89 |

Table 15: OOD detection results on the MIDOG dataset. Because the MIDOG sub-datasets are indexed by numeric codes rather than descriptive names, we denote each OOD sub-dataset using its code in the first row. These sub-datasets differ from the ID data due to variations in staining protocols, imaging devices, acquisition centers, tissue preparation, and underlying cell types. The best and second-best results are highlighted in bold and underlined, respectively

| Method | Performance | |
|---|---|---|
| | FPR↓ | AUC↑ |
| SCALE (RNet50) (Xu et al., 2023) | 25.29 | 86.68 |
| OTOD (RNet50) (Gao et al., 2025) | 30.57 | 83.81 |
| NAC-UE (RNet50) (Liu et al., 2023b) | 25.74 | 86.25 |
| CATEX (CLIP) (Liu et al., 2023a) | 29.33 | 82.19 |
| ViM (ViT)(Wang et al., 2022) | 23.17 | 90.18 |
| Ours (RNet50) | 24.63 | 88.46 |
| Ours (ViT) | 21.8 | 92.51 |

Table 16: OOD detection results on the SpaceNet-8 dataset. The non-flooded dataset from two cities is used as ID, whereas the flooded dataset is used as OOD. The best and second-best results are highlighted in bold and underlined, respectively.

| Method | SVHN | | Textures | | LSUN-C | | LSUN-R | | iSUN | | Places365 | | Average | |
|---|---|---|---|---|---|---|---|---|---|---|---|---|---|---|
| | FPR↓ | AUC↑ | FPR↓ | AUC↑ | FPR↓ | AUC↑ | FPR↓ | AUC↑ | FPR↓ | AUC↑ | FPR↓ | AUC↑ | FPR↓ | AUC↑ |
| Ours (UAT) | 6.98 ± 1.2 | 98.84 ± 0.3 | 15.40 ± 1.4 | 97.28 ± 0.2 | 0.15 ± 0.0 | 99.95 ± 0.0 | 10.80 ± 4.3 | 97.88 ± 0.4 | 6.42 ± 3.3 | 98.23 ± 0.2 | 49.32 ± 1.5 | 91.25 ± 0.5 | 14.85 ± 1.4 | 97.24 ± 0.3 |
| Ours (UAT + CCFN) | 7.25 ± 0.9 | 98.72 ± 0.2 | 15.13 ± 1.4 | 97.34 ± 0.2 | 0.37 ± 0.1 | 99.91 ± 0.0 | 6.51 ± 4.6 | 98.77 ± 0.8 | 5.09 ± 3.3 | 99.05 ± 0.5 | 38.92 ± 1.0 | 93.04 ± 0.2 | 12.21 ± 1.3 | 97.80 ± 0.2 |

Table 17: The standard deviation values for UAT and UAT+CCFN, as reported in Table 1 of our main paper, are provided here.

| Method | SVHN | | Textures | | LSUN-C | | LSUN-R | | iSUN | | Places365 | | Average | |
|---|---|---|---|---|---|---|---|---|---|---|---|---|---|---|
| | FPR↓ | AUC↑ | FPR↓ | AUC↑ | FPR↓ | AUC↑ | FPR↓ | AUC↑ | FPR↓ | AUC↑ | FPR↓ | AUC↑ | FPR↓ | AUC↑ |
| MSP (Hendrycks & Gimpel, 2016) | 42.10 | 91.85 | 53.30 | 87.45 | 24.85 | 96.37 | 37.81 | 93.71 | 40.11 | 93.73 | 50.73 | 88.58 | 41.49 | 91.84 |
| EN (Liu et al., 2020) | 33.11 | 90.54 | 46.06 | 85.09 | 5.86 | 98.76 | 22.68 | 94.90 | 25.12 | 94.17 | 39.08 | 88.50 | 28.65 | 91.99 |
| EN+REACT (Sun et al., 2021) | 98.31 | 39.94 | 91.85 | 60.80 | 96.76 | 57.11 | 80.15 | 77.63 | 79.48 | 79.48 | 77.98 | 73.29 | 65.78 | 86.22 |
| EN+DICE (Sun & Li, 2022) | 37.84 | 86.99 | 50.77 | 79.70 | 2.54 | 99.43 | 26.30 | 92.89 | 28.30 | 92.89 | 43.46 | 84.65 | 31.53 | 89.30 |
| DML+ (Zhang & Xiang, 2023) | 16.43 | 95.62 | 20.31 | 94.81 | 2.5 | 99.03 | 24.59 | 96.14 | 20.74 | 96.28 | 40.26 | 83.77 | 19.8 | 94.27 |
| OTOD (Song et al., 2024) | 20.45 | 95.21 | 30.87 | 94.56 | 6.75 | 98.56 | 33.32 | 94.21 | 31.34 | 94.39 | 40.27 | 92.63 | 27.17 | 94.93 |
| ODIN (Gao et al., 2025) | 37.08 | 88.36 | 47.58 | 82.85 | 6.14 | 98.65 | 20.51 | 95.04 | 22.95 | 94.22 | 41.03 | 86.57 | 29.22 | 90.95 |
| RFW (Liang et al., 2017) | 21.05 | 95.05 | 43.88 | 86.16 | 5.30 | 98.76 | 19.45 | 95.97 | 22.25 | 95.36 | 31.23 | 91.63 | 23.86 | 93.82 |
| FN(Yu et al., 2023) | 3.83 | 99.18 | 14.23 | 97.06 | 0.32 | 99.81 | 8.13 | 98.32 | 5.98 | 98.71 | 48.69 | 90.91 | 13.53 | 97.33 |
| PRO (Chen et al., 2025) | 13.31 | 96.79 | 21.31 | 95.03 | 4.11 | 99.06 | 23.84 | 96.51 | 21.98 | 95.17 | 21.31 | 95.10 | 17.64 | 96.28 |
| Ours (UAT + CCFN) | 4.39 ± 0.9 | 99.17 ± 0.2 | 8.93 ± 0.1 | 98.51 ± 0.0 | 0.24 ± 0.0 | 99.92 ± 0.0 | 4.05 ± 0.7 | 99.17 ± 0.1 | 3.38 ± 0.3 | 99.24 ± 0.1 | 20.41 ± 0.5 | 95.83 ± 0.1 | 6.90 ± 0.1 | 98.64 ± 0.0 |

Table 18: Detailed experimental results corresponding to Table 2 in our main paper are provided. The ID dataset was CIFAR-10, the backbone is WRN-28-10, and the OOD datasets are listed in the first row of each table. For clarity, the best results are highlighted in bold, and the second-best results are underlined.

| Method | SVHN FPR↓ | AUC↑ | Textures FPR↓ | AUC↑ | LSUN-C FPR↓ | AUC↑ | LSUN-R FPR↓ | AUC↑ | iSUN FPR↓ | AUC↑ | Places365 FPR↓ | AUC↑ | Average FPR↓ | AUC↑ |
|---|---|---|---|---|---|---|---|---|---|---|---|---|---|---|
| MSP (Hendrycks & Gimpel, 2016) | 68.07 | 90.02 | 63.86 | 89.37 | 46.63 | 93.73 | 70.19 | 86.29 | 71.81 | 85.71 | 68.08 | 87.25 | 64.77 | 88.73 |
| EN (Liu et al., 2020) | 53.11 | 92.26 | 47.04 | 92.08 | 18.51 | 97.20 | 53.02 | 89.58 | 55.39 | 88.97 | 51.67 | **89.95** | 46.46 | 91.67 |
| EN+REACT (Sun et al., 2021) | 58.16 | 83.28 | 51.73 | 87.47 | 23.40 | 94.77 | 47.19 | 89.68 | 51.30 | 87.39 | 50.47 | 87.39 | 47.15 | 88.44 |
| EN+DICE (Sun & Li, 2022) | 47.81 | 93.27 | 50.95 | 91.77 | 16.73 | 97.06 | 64.26 | 87.83 | 65.83 | 87.83 | 59.23 | 88.53 | 50.80 | 90.98 |
| DML+ (Zhang & Xiang, 2023) | 68.62 | 91.30 | 33.3 | 92.5 | 19.46 | 95.49 | 79.27 | 82.70 | 59.34 | 88.14 | 66.59 | 86.88 | 54.43 | 89.17 |
| RFW (Song et al., 2024) | 67.81 | 88.60 | 56.12 | 87.80 | 18.63 | 96.49 | **46.34** | 88.81 | 51.59 | 86.54 | 49.37 | 88.44 | 48.31 | 89.45 |
| OTOD (Gao et al., 2025) | 46.85 | 86.64 | 44.91 | 81.39 | 20.87 | 88.67 | 56.04 | 89.05 | 55.47 | 90.11 | 65.11 | 86.64 | 48.21 | 87.08 |
| ODIN (Liang et al., 2017) | 53.86 | 92.23 | 48.09 | 91.94 | 19.95 | 97.01 | 54.29 | 89.47 | 56.61 | 88.57 | 52.34 | 89.86 | 47.52 | 91.56 |
| FN (Yu et al., 2023) | 8.84 | 98.24 | **24.62** | 95.11 | 3.38 | 99.36 | 71.17 | 83.12 | 62.80 | 86.05 | 65.25 | 85.20 | 39.34 | 91.18 |
| PRO (Chen et al., 2025) | 37.09 | 97.49 | 58.33 | 84.27 | 14.08 | 95.24 | 48.33 | 89.72 | 55.07 | 87.83 | 65.49 | 84.28 | 46.40 | 89.81 |
| Ours (UAT + CCFN) | **8.78** ± 0.8 | 98.20 ± 0.1 | 27.63 ± 0.4 | 95.09 ± 0.0 | 2.86 ± 0.6 | 99.43 ± 0.1 | 60.94 ± 2.1 | 88.65 ± 0.2 | 56.65 ± 1.4 | 90.06 ± 0.3 | 56.41 ± 3.6 | 88.29 ± 0.9 | **35.55** ± 0.2 | **93.29** ± 0.0 |

Table 19: Detailed experimental results corresponding to Table 2 in our main paper are provided. The ID dataset was CIFAR-10, the backbone is VGG11, and the OOD datasets are listed in the first row of each table. For clarity, the best results are highlighted in bold, and the second-best results are underlined.

| Method | SVHN FPR↓ | AUC↑ | Textures FPR↓ | AUC↑ | LSUN-C FPR↓ | AUC↑ | LSUN-R FPR↓ | AUC↑ | iSUN FPR↓ | AUC↑ | Places365 FPR↓ | AUC↑ | Average FPR↓ | AUC↑ |
|---|---|---|---|---|---|---|---|---|---|---|---|---|---|---|
| MSP (Hendrycks & Gimpel, 2016) | 70.15 | 78.30 | 65.69 | 66.86 | 38.52 | 85.53 | 78.45 | 74.17 | 84.24 | 67.68 | 91.62 | 55.42 | 71.45 | 71.33 |
| EN (Liu et al., 2020) | 66.38 | 81.84 | 63.30 | 68.64 | 27.90 | 90.06 | 78.39 | 75.30 | 82.85 | 70.36 | 91.17 | 56.27 | 68.33 | 73.75 |
| EN+REACT (Sun et al., 2021) | 69.10 | 77.96 | 66.08 | 65.86 | 36.07 | 85.90 | 78.87 | 73.51 | 84.75 | 67.02 | 90.41 | 56.33 | 70.88 | 71.10 |
| EN+DICE (Sun & Li, 2022) | 68.81 | 78.20 | 65.96 | 65.99 | 34.86 | 86.36 | 78.77 | 73.68 | 84.63 | 67.28 | 90.25 | 56.46 | 70.55 | 71.33 |
| DML+ (Zhang & Xiang, 2023) | 72.63 | 76.05 | 69.06 | 65.02 | 48.44 | 81.82 | 80.33 | 72.10 | 86.21 | 65.11 | 91.36 | 55.40 | 74.67 | 69.25 |
| RFW (Song et al., 2024) | 85.19 | 76.29 | 89.96 | 66.27 | 49.46 | 90.14 | 74.81 | 81.89 | 79.56 | 78.61 | 86.34 | 69.85 | 77.55 | 77.18 |
| OTOD (Gao et al., 2025) | 59.78 | 84.55 | 57.27 | 73.18 | 11.63 | 96.55 | 72.11 | 79.15 | 74.52 | 77.30 | 87.35 | 61.40 | 60.44 | 78.69 |
| ODIN (Liang et al., 2017) | 61.39 | 83.49 | 59.38 | 71.04 | 15.41 | 94.96 | 75.83 | 77.93 | 77.29 | 74.87 | 88.06 | 60.20 | 62.89 | 77.08 |
| FN (Yu et al., 2023) | **23.34** | 92.75 | 68.62 | 71.47 | 19.46 | 93.88 | 79.27 | 81.08 | 89.34 | 72.94 | 81.59 | 72.42 | 60.27 | 80.76 |
| PRO (Chen et al., 2025) | 47.49 | 84.82 | 64.27 | 77.21 | 59.49 | 88.08 | 71.25 | 82.84 | 61.07 | 89.85 | 56.95 | 89.82 | 60.09 | 85.44 |
| Ours(UAT + CCFN) | 28.72 ± 1.4 | **94.06** ± 0.4 | 39.75 ± 0.6 | 87.64 ± 0.5 | 7.92 ± 4.4 | 98.49 ± 0.7 | 54.23 ± 5.7 | 89.38 ± 1.1 | 48.16 ± 5.7 | 90.79 ± 1.0 | 91.83 ± 0.6 | 61.12 ± 2.7 | **45.10** ± 1.6 | 86.91 ± 0.6 |

Table 20: Detailed experimental results corresponding to Table 2 in our main paper are provided. The ID dataset was CIFAR-100, the backbone is ResNet-18, and the OOD datasets are listed in the first row of each table. For clarity, the best results are highlighted in bold, and the second-best results are underlined.

| Method | iNaturalist FPR↓ | AUC↑ | SUN FPR↓ | AUC↑ | PLACES FPR↓ | AUC↑ | Textures FPR↓ | AUC↑ | Average FPR↓ | AUC↑ |
|---|---|---|---|---|---|---|---|---|---|---|
| Ours(UAT) | 18.58 ± 0.8 | 95.84 ± 0.2 | 36.81 ± 1.1 | 91.42 ± 0.3 | 44.94 ± 1.7 | 88.62 ± 0.4 | 86.39 ± 0.9 | 59.75 ± 0.8 | 46.68 ± 0.6 | 83.91 ± 0.2 |
| Ours(UAT + CCFN) | 18.46 ± 0.7 | 96.11 ± 0.1 | 34.75 ± 1.0 | 91.93 ± 0.4 | 43.79 ± 1.5 | 89.26 ± 0.4 | 85.77 ± 0.7 | 60.23 ± 0.7 | 45.69 ± 0.4 | 84.39 ± 0.2 |

Table 21: Standard Deviation values of UAT and (UAT + CCFN) in Table 3 of our main paper are provided.

| Method | SVHN FPR↓ | AUC↑ | Textures FPR↓ | AUC↑ | LSUN-C FPR↓ | AUC↑ | LSUN-R FPR↓ | AUC↑ | iSUN FPR↓ | AUC↑ | Places365 FPR↓ | AUC↑ | Average FPR↓ | AUC↑ |
|---|---|---|---|---|---|---|---|---|---|---|---|---|---|---|
| $f_{std}(x)$ | 9.07 ± 0.6 | 98.45 ± 0.1 | 36.15 ± 3.4 | 91.15 ± 0.7 | 0.27 ± 0.0 | 99.88 ± 0.0 | 31.28 ± 2.8 | 95.76 ± 0.7 | 32.31 ± 3.6 | 93.77 ± 0.6 | 60.80 ± 1.0 | 85.00 ± 0.7 | 28.31 ± 1.7 | 94.00 ± 0.2 |
| $f_{std}(x + L_{adv})$ | 13.27 ± 0.8 | 97.61 ± 0.2 | 37.68 ± 3.2 | 90.72 ± 0.5 | 0.49 ± 0.2 | 99.73 ± 0.1 | 36.95 ± 0.6 | 94.10 ± 1.5 | 35.81 ± 2.8 | 91.89 ± 1.5 | 63.60 ± 2.0 | 84.77 ± 0.4 | 31.30 ± 1.2 | 93.14 ± 0.4 |
| $f_{uat}(x)$ | 7.46 ± 1.1 | 98.25 ± 0.3 | 16.32 ± 1.1 | 97.18 ± 0.1 | 0.22 ± 0.0 | 99.91 ± 0.0 | 12.77 ± 4.1 | 97.90 ± 0.5 | 7.94 ± 3.1 | 97.50 ± 0.2 | 49.56 ± 1.5 | 91.11 ± 0.4 | 15.71 ± 1.3 | 96.98 ± 0.3 |
| $f_{uat}(x + L_{adv})$ | 6.98 ± 1.2 | 98.84 ± 0.3 | 15.40 ± 1.4 | 97.28 ± 0.2 | 0.15 ± 0.0 | 99.95 ± 0.0 | 10.80 ± 4.3 | 97.88 ± 0.4 | 6.42 ± 3.3 | 98.23 ± 0.2 | 49.32 ± 1.5 | 91.25 ± 0.5 | 14.85 ± 1.4 | 97.24 ± 0.3 |

Table 22: Detailed experimental results of OOD detection metrics corresponding to Table 4 in our main paper are provided. The ID dataset was CIFAR-10, the backbone is ResNet-18, and the OOD datasets are listed in the first row of each table.

| Method | SVHN | Textures | LSUN-C | LSUN-R | iSUN | Places365 | Avg |
|---|---|---|---|---|---|---|---|
| $f_{std}(x)$ | 1.721 | 1.302 | 2.291 | 1.493 | 1.568 | 1.294 | 1.612 |
| $f_{std}(x + L_{adv})$ | 1.572 | 1.206 | 2.112 | 1.457 | 1.502 | 1.258 | 1.518 |
| $f_{uat}(x)$ | 1.846 | 1.834 | 2.392 | 1.886 | 2.000 | 1.477 | 1.906 |
| $f_{uat}(x + L_{adv})$ | 1.887 | 1.948 | 2.429 | 1.961 | 2.020 | 1.481 | 1.954 |

Table 23: Detailed experimental results of norm ratio corresponding to Table 4 in our main paper are provided. The ID dataset was CIFAR-10, backbone is ResNet-18 and the OOD datasets are listed in the first row of each table.

