# OpenReview forum: "Actively Enlarging Feature Norms with Universal Adversarial Training and Channel Calibration for Superior OOD Detection"
_ICLR.cc/2026/Conference — ICLR 2026 Conference Withdrawn Submission_

### Official Review · Reviewer_Zdr8 · 2025-10-31

**Soundness:** 3
**Presentation:** 2
**Contribution:** 2
**Rating:** 4
**Confidence:** 3

**Summary:**

The paper introduces a novel framework aimed at improving out-of-distribution (OOD) detection by focusing on the feature norm gap between in-distribution (ID) and OOD data. The authors propose Universal Adversarial Training (UAT) combined with a Channel-Calibrated Feature Norm (CCFN) scoring mechanism. UAT leverages a universal adversarial map (UAM) to enhance class separation during training, while CCFN improves the performance at inference by emphasizing informative features and suppressing irrelevant ones. The proposed method demonstrates significant improvements in OOD detection across several benchmarks, including CIFAR-10, CIFAR-100, and ImageNet, achieving state-of-the-art results.

**Strengths:**

1. The combination of UAT and CCFN is a novel method that fills a significant gap in OOD detection methods.

2. This method has a solid theoretical foundation and is validated through comprehensive experiments.

3. This paper provides a clear description of the method, with sufficient detail in both the theoretical and experimental sections.

**Weaknesses:**

1. While the theoretical framework is solid, the section involving the proofs (especially Lemmas 1 and 2) may be difficult for some readers to follow without sufficient context or explanation.

2. The paper could balance the theoretical content with more practical insights or intuitive examples to make it more accessible for a broader audience.

3. While the experiments show clear advantages, additional comparisons with more recent OOD detection techniques, such as [1] and [2], could further solidify the proposed method’s superiority.

4. The paper does not provide a link to an anonymized code repository, which could hinder future efforts for replicating or building upon the work. Open-sourcing the implementation would not only facilitate reproducibility but also contribute to the broader research community.

5. While the method shows strong performance on image datasets, the paper does not provide sufficient analysis on how well the approach generalizes to other domains, such as time-series or textual data. Given the importance of OOD detection in various fields, exploring the adaptability of the method beyond images would be valuable.

References

[1] Xu, K., Chen, R., Franchi, G., & Yao, A. (2023). Scaling for training time and post-hoc out-of-distribution detection enhancement. arXiv preprint arXiv:2310.00227.

[2] Liu, Y., Tian, C. X., Li, H., Ma, L., & Wang, S. (2023). Neuron activation coverage: Rethinking out-of-distribution detection and generalization. arXiv preprint arXiv:2306.02879.

**Questions:**

1. Does the proposed method generalize to other domains beyond image data? Could it be applied to, for example, time-series or text data for OOD detection?

2. Could the results in Table 6 be further improved with different strategies for updating the adversarial map during training, or is the chosen frequency optimal across all datasets?

---

> ### Author Response · Authors · 2025-11-24
> **We appreciate the reviewer’s(Zdr8) insightful comments. Below, we address the questions raised by the reviewer. (Part 1)**
>
> # Reviewer Zdr8
>
> **Q1:** While the theoretical framework is solid, proofs (especially Lemmas 1 and 2) may be difficult to follow without sufficient context. The paper could balance theory with intuitive examples.
>
> **A1:** We appreciate the reviewer's suggestion to provide more intuition alongside the formal proofs. In this rebuttal, we include an additional visualization (Fig. 1 of the revised paper) that offers an intuitive illustration of how the UAM influences the decision boundary between classes. As the model is trained to be robust against a set of universal perturbations, the intra-class features become more compact and aligned, while the means of different classes move farther apart. This geometric adjustment makes the intra-class distributions more compact, which in turn reduces the likelihood that OOD samples fall near the ID feature regions. As a result, OOD samples are pushed farther away from the ID manifold and become more distinguishable.
>
> Below, we summarize the intuition behind Lemmas 1 and 2.
>
> For intra-class compactness (Lemma 1), the requirement that the classifier remain robust against a universal perturbation encourages the model to smooth the decision boundary and push it outward from the data manifold. Geometrically, this forces the class-conditional feature distributions to become more convex and aligned. The optimization therefore drives features *f_θ(x)* to move closer to their corresponding class prototypes *W_k* in order to maintain consistent predictions under the perturbation. This results in reduced angular variance within each class (i.e., the angle between *f_θ(x)* and *W_k*), which corresponds to a tighter, more compact intra-class cluster.
>
> For inter-class separability (Lemma 2), the universal perturbation implicitly constrains the feature norms, so the model must enlarge the classification margin *z_k - z_j* through angular separation rather than feature scaling. As shown in Appendix A.2 (Eq. 12), the robust margin is directly proportional to the geometric term *(1 - \cos(\beta_{kj})*, where *\beta_{kj}* is the angle between two class prototypes *W_k* and *W_j*. Maximizing this margin, therefore, requires increasing the angular distance between prototypes. This angular expansion translates into greater Euclidean separation between class means and produces a more discriminative feature space under adversarial robustness.
>
> **Q2:** Additional comparisons with more recent OOD detection techniques such as [5] and [7] would strengthen the work.
>
> **A2:** In this rebuttal, we have compared these two baselines in Table 3 of the revised paper and in the new experimental results at Tables 1 and 2. Particularly, these new comparative studies were conducted on more realistic datasets: MIDOG [1] and SpaceNet-8 [2] datasets.
>
> -   We thank the reviewer for the suggestion. We have extended our method to a ViT backbone and evaluated its performance on the MIDOG [1] and SpaceNet-8 [2] datasets.The MIDOG (Mitosis Domain Generalization) dataset is a digital pathology dataset, focusing on mitotic figure detection in H&E-stained breast cancer tissue. The MIDOG dataset is divided into several sub-datasets, and each sub-dataset represents a distinct domain with variations in imaging conditions. The domains differ in scanner type, staining protocol, image resolution, acquisition center, and overall visual style. Specifically, datasets 1a-c are sourced from different scanners within the same institution, datasets 2-7 introduce differences in staining protocols, imaging devices, acquisition centers, tissue preparation, and different cancer types from human and canine species. In our evaluation, we used 1a-c as our ID, whereas the rest were used as OOD.
>
> -   SpaceNet-8 is a large-scale remote sensing dataset for flood detection, containing multi-spectral satellite imagery from several geographic regions. In our experiment on SpaceNet-8, we treat non-flooded images from two cities as ID, and use flooded images as OOD samples to evaluate the model's ability to detect distributional shift.

---

> ### Author Response · Authors · 2025-11-24
> **We appreciate the reviewer’s(Zdr8) insightful comments. Below, we address the questions raised by the reviewer. (Part 2)**
>
> #### Robustness across Backbone Architectures (ViT).
>
> To demonstrate the generality of our approach, we extended the UAT framework to the Vision Transformer (ViT) backbone. The core logic of UAT remains applicable, but we adapt the OOD scoring mechanism to the ViT's feature space, following established practices for OOD detection in ViT. Specifically, after training ViT with UAT, we extract the penultimate-layer features and apply a channel-wise gating mechanism (max activation) to emphasize the most informative dimensions for inference. To calculate the norm values, we follow the ViM [3] formulation. We compute the feature center using the pseudo-inverse of the classifier weights and estimate the covariance of centered features. The smallest-variance eigenvectors form an orthogonal subspace, and projecting features onto this subspace yields the orthogonal projection norm, which quantifies deviation from the ID feature manifold.
>
> **Experiment Setup.** To ensure a fair comparison, all baseline methods were run using their official repositories. Among these baselines, CATEX[4] is a CLIP-based approach pretrained on ImageNet. To adapt CATEX to MIDOG and SpaceNet-8, we followed its original design and constructed domain- and label-specific textual prompts, then finetuned the model to identify visual patterns that deviate from the expected distribution, such as unfamiliar cellular textures or image domains unrelated to the training data.
>
> **Results.** As shown in Tables 1 and 2, our method consistently yields substantial improvements over all competing approaches on both datasets. Notably, CATEX performs significantly worse than the other baselines, which we believe is due to the mismatch between the general-purpose semantics encoded in the ImageNet-pretrained CLIP backbone and the highly specialized, fine-grained visual cues required for medical pathology (MIDOG) and satellite imagery (SpaceNet-8). These results demonstrate that UAT remains effective across both CNN and non-CNN architectures and that it generalizes well to real-world distribution shifts that go far beyond conventional OOD detection benchmarks. We have added these new results to Section A.12 of the revised manuscript for completeness.
>
> | Method               | 2 FPR↓ | 2 AUC↑ | 3 FPR↓ | 3 AUC↑ | 4 FPR↓ | 4 AUC↑ | 5 FPR↓ | 5 AUC↑ | 6a FPR↓ | 6a AUC↑ | 6b FPR↓ | 6b AUC↑ | 7 FPR↓ | 7 AUC↑ | Avg FPR↓ | Avg AUC↑ |
> | :------------------- | :----: | :----: | :----: | :----: | :----: | :----: | :----: | :----: | :-----: | :-----: | :-----: | :-----: | :----: | :----: | :------: | :------: |
> | SCALE (ResNet50) [5]      | 67.60  | 72.53  | 67.51  | 72.43  | 28.20  | 85.78  | 54.84  | 78.85  |  66.09  |  69.06  |  69.88  |  78.29  | 55.24  | 78.15  |  58.48   |  76.44   |
> | OTOD (ResNet50) [6]       | 64.61  | 69.28  | 66.16  | 67.07  | 22.59  | 92.65  | 63.31  | 66.38  |  59.38  |  69.77  |  58.38  |  80.70  | 61.17  | 71.46  |  56.51   |  73.90   |
> | NAC-UE (ResNet50) [7]     | 59.08  | 75.91  | 59.61  | 77.80  | 30.14  | 82.46  | 56.23  | 71.79  |  57.99  |  76.61  |  61.88  | 80.96| 58.95  | 75.28  |  54.84   |  77.26   |
> | CATEX (CLIP) [4]         | 75.41  | 70.87  | 71.22  | 74.65  | 20.30  | 92.23  | 70.60  | 68.60  |  60.01  |  75.66  |  86.18  |  73.74  | 68.29  | 75.53  |  64.57   |  75.90   |
> | ViM (ViT) [3]           | 55.37| 77.14| **48.39** | 75.56  | 23.73  | 92.12  | **48.24** | 79.59  |  **41.04** | 83.39| 89.42  | 63.14  | 50.90| 76.25  |  51.01   |  78.17   |
> | Ours (ResNet50)        | 58.86  | 76.57  | 52.16  | 81.49| 18.91| 93.42| 48.52| 81.02| 48.32  | 79.49  | 67.91| 73.38  | **48.03** | **85.74** | 48.96| 80.56|
> | Ours (ViT)           | **51.20** | **81.29** | 53.17| **81.91** | **17.91** | **94.95** | 49.05  | **82.21** | 44.93| **85.80** | **60.14** | **80.54** | 53.89  | 80.50| **47.18** | **83.89** |
>
> **Table 1:**: OOD detection results on the MIDOG dataset. Because the MIDOG sub-datasets are indexed by numeric codes rather than descriptive names, we denote each OOD sub-dataset using its code in the first row. These sub-datasets differ from the ID data due to variations in staining protocols, imaging devices, acquisition centers, tissue preparation, and underlying cell types. The best and second-best results are highlighted in bold and underlined, respectively.
>
>
> | Method               | FPR↓ | AUC↑ |
> | :------------------- | :--: | :--: |
> | SCALE (ResNet50) [5]       | 25.29 | 86.68 |
> | OTOD (ResNet50) [6]       | 30.57 | 83.81 |
> | NAC-UE (ResNet50) [7]     | 25.74 | 86.25 |
> | CATEX (CLIP) [4]        | 29.33 | 82.19 |
> | ViM (ViT) [3]           | 23.17| 90.18|
> | Ours (ResNet50)        | 24.63 | 88.46 |
> | Ours (ViT)           | **21.8** | **92.51** |
>
> **Table 2:**: OOD detection results on the SpaceNet-8 dataset. The non-flooded datasets from two cities are used as ID, whereas the flooded dataset is used as OOD. The best and second-best results are highlighted in bold and underlined, respectively

---

> > ### Author Response · Authors · 2025-11-24
> > **We appreciate the reviewer’s(Zdr8) insightful comments. Below, we address the questions raised by the reviewer. (Part 3)**
> >
> > **Q3:** The paper does not provide a link to an anonymized code repository.
> >
> > **A3:** We have included our code in the supplementary material. We will release the code soon after publication.
> >
> > **Q4:** The method's generalization to other domains (e.g., time-series, text) is not discussed.
> >
> > **A4:** We appreciate the reviewer's thoughtful and forward-looking question. Although our study focuses on visual data, the underlying ideas of UAT are not limited to images. The UAM operates through small continuous perturbations in the input space and could, in principle, be applied to time-series data, where the input domain is also continuous. For text, despite its discrete nature, similar perturbation-based regularization could be introduced by working in a continuous latent
> > space, such as the embeddings produced by Sentence-VAE [8], CLIP, or Transformer encoders. In such settings, UAM-style perturbations and channel-calibrated norm scoring could be applied directly to the continuous embedding vectors.
> >
> > Regarding CCFN, the mechanism is model-agnostic and is designed to rescale multi-channel feature representations by suppressing irrelevant or noisy channels. Since feature maps or hidden states in many non-visual domains also exhibit a multi-channel structure, the same recalibration strategy can naturally extend to modalities such as time series and text. We view exploring these directions as promising future work.
> >
> > **Q5:** Could the results in Table 6 be further improved with different strategies for updating the adversarial map during training, or is the chosen frequency optimal across all datasets?
> >
> > **A5:** We appreciate the reviewer's question regarding the generality of the UAM update frequency *τ*. In our experiments, the setting *τ = 50* consistently produced strong and stable improvements across all tested ID datasets and backbone architectures. This suggests that *τ = 50* is a robust default choice for our UAT framework under diverse conditions.
> >
> > That said, we agree that the theoretically optimal update frequency may depend on factors such as dataset complexity or model capacity. Exploring adaptive or data-driven strategies for determining *τ* is indeed a promising direction. We consider this an excellent avenue for future work and expect that dynamically adjusting the update frequency during training could further fine-tune performance in particularly challenging or specialized scenarios.
> >
> >
> > **Refernce:**
> >
> > [1] Marc Aubreville, Frauke Wilm, Nikolas Stathonikos, Katharina Breininger, Taryn A Donovan, Samir Jabari, Mitko Veta, Jonathan Ganz, Jonas Ammeling, Paul J van Diest, et al. A comprehensive multi-domain dataset for mitotic figure detection. Scientific data, 10(1):484, 2023.
> >
> > [2] Ronny Hänsch, Jacob Arndt, Dalton Lunga, Matthew Gibb, Tyler Pedelose, Arnold Boedihardjo, Desiree Petrie, and Todd M Bacastow. Spacenet 8-the detection of flooded roads and buildings. In Proceedings of the IEEE/CVF conference on computer vision and pattern recognition, pp. 1472–1480, 2022.
> >
> > [3] Xiaoyun Wang, Weitang Liu, Shiyu Liang, and Yixuan Li. Vim: Out-of-distribution with virtual-logit matching. In Proceedings of the IEEE/CVF Conference on Computer Vision and Pattern Recognition, pp. 4921–4930, 2022.
> >
> > [4] Kai Liu, Zhihang Fu, Chao Chen, Sheng Jin, Ze Chen, Mingyuan Tao, Rongxin Jiang, and Jieping Ye. Category-extensible out-of-distribution detection via hierarchical context descriptions. Advances in Neural Information Processing Systems, 36:33241–33261, 2023a.
> >
> > [5] Kai Xu, Rongyu Chen, Gianni Franchi, and Angela Yao. Scaling for training time and post-hoc out-of-distribution detection enhancement. arXiv preprint arXiv:2310.00227, 2023.
> >
> > [6] Heng Gao, Zhuolin He, and Jian Pu. Detecting ood samples via optimal transport scoring function. In ICASSP 2025-2025 IEEE International Conference on Acoustics, Speech and Signal Processing (ICASSP), pp. 1–5. IEEE, 2025.
> >
> > [7] Yibing Liu, Chris Xing Tian, Haoliang Li, Lei Ma, and Shiqi Wang. Neuron activation coverage: Rethinking out-of-distribution detection and generalization. arXiv preprint arXiv:2306.02879, 2023b.
> >
> > [8] Hongjun An, Yifan Chen, Zhe Sun, and Xuelong Li. Sentencevae: Enable next-sentence prediction for large language models with faster speed, higher accuracy and longer context. arXiv preprint arXiv:2408.00655, 2024

---

> ### Author Response · Authors · 2025-11-27
> **Follow-up on Rebuttal for Paper ID 6514**
>
> Dear reviewer Zdr8,
>
> We have addressed all the raised questions and concerns comprehensively. We hope that a quick review of our rebuttal will confirm that your valuable feedback has been thoroughly incorporated.
>
> If you have any remaining questions or require further clarification on our rebuttal or the revised paper content, please do not hesitate to reach out. If our revisions and explanations have resolved your concerns, we would be grateful for your support and encourage you to reflect this in your final score.
>
> Thank you.
> The authors

---

### Official Review · Reviewer_roTf · 2025-11-01

**Soundness:** 2
**Presentation:** 3
**Contribution:** 2
**Rating:** 4
**Confidence:** 4

**Summary:**

This paper introduces a new method for OOD detection, which actively enlarges the feature norm gap between ID and OOD data through Universal Adversarial Training (UAT) and Channel-Calibrated Feature Norm (CCFN). UAT employs a learnable universal adversarial map (UAM) as a regularizer during training; CCFN optimizes the OOD score at inference time by re-weighting channel activations. The authors demonstrate the superiority of their method over existing techniques through extensive experiments on CIFAR-10, CIFAR-100, and ImageNet.

**Strengths:**

1. The method achieves state-of-the-art or near-SOTA performance on all reported benchmarks, with particularly significant reductions in the FPR95 metric.
2. Compared to traditional per-sample adversarial training, UAT only needs to optimize a single, global UAM, which is computationally more efficient. The appendix also shows this approach is more effective for OOD detection.
3. The paper provides an in-depth analysis of the method's key components, including the impact of UAT on the norm ratio, the UAM update frequency, and the perturbation location, all of which strengthen the study's claims.

**Weaknesses:**

1.  Although the proposed method improves OOD detection performance, the paper lacks an analysis of its computational efficiency. The introduction of the learnable universal adversarial map (UAM) and the channel-calibrated feature norm (CCFN) scoring mechanism likely adds computational overhead. A discussion on the method's efficiency, including training time and inference speed, would provide a clearer understanding of its practical applicability in resource-constrained environments.
2.  The proposed adversarial training method introduces a universal adversarial map to regularize the model. However, this may lead to overfitting, especially if the model becomes too adapted to this specific adversarial perturbation.
3.  The OOD for the newly trained model is no longer the OOD for the original model, it's just easier to detect on the new model. If I want to detect the OOD for the original model, this is not applicable. In many cases, OOD is not only for defense, but also a means to adjust the model, thus losing an important tool.
4.  How can this method be extended beyond classification models? OOD detection is not limited to classification models, is it?

**Questions:**

1. Can the authors provide specific data on the computational overhead introduced by UAT (vs. standard training) and CCFN (vs. standard L2 norm scoring)? For example, by what percentage does training time increase, and what is the added inference latency (ms/image) on ResNet-50 for ImageNet?
2. If the model overfits to $L_{adv}$, does its robustness against other types of unseen adversarial perturbations or data corruptions decrease compared to a standard-trained model?
3. If my goal is to evaluate the OOD robustness of an existing, pre-trained model that cannot be retrained, is the proposed method still applicable?
4. Have the authors considered how the UAT framework could be extended to OOD detection in non-classification tasks, such as object detection or semantic segmentation, where the loss functions and feature norm definitions might need adaptation?

---

> ### Author Response · Authors · 2025-11-24
> **We appreciate the reviewer’s(roTf) insightful comments. Below, we address the questions raised by the reviewer.**
>
> # Reviewer roTf
>
>
>
> **Q1:** The paper lacks an analysis of computational efficiency. The introduction of UAM and CCFN likely adds overhead. Please provide training and inference statistics.
>
> **A1:** The overhead is minimal. For training, the cost is very low because we update the universal adversarial map (UAM) infrequently (*τ=50*, Table 6 of the revised paper). For instance, 100 epochs of UAT training on ImageNet (ResNet-50) takes 27 hours 10 minutes, which is only about 5.8% longer than the 25 hours 40 minutes for standard training. For inference, the channel-calibrated feature norm (CCFN) introduces negligible latency. The average time per image increases from 0.0123 seconds (standard *L2* norm) to just 0.0127 seconds (CCFN score), confirming a minimal increase of about 3.3%. Our method is highly efficient for practical deployment. [We also added this information in Section A.11 of the revised manuscript.
>
> **Q2:** The proposed adversarial training introduces a universal adversarial map. This may lead to overfitting if the model becomes too adapted to a specific perturbation. If the model overfits to the map, does its robustness to unseen perturbations or corruptions degrade?
>
> **A2:** We argue that the UAM is dynamically updated every *τ=50* iterations (as analyzed in Table 6), which is essential to prevent over-adaptation to any single static pattern. Experiment results also indicated that UAT acts as a strong generalizer rather than a source of overfitting. Table 5 confirms UAT consistently improves ID test accuracy in several backbones (e.g., ResNet-18 accuracy improves from 94.0% to 95.0%), demonstrating enhanced generalization.
>
> **Q3:** If my goal is to evaluate the OOD robustness of an existing pretrained model that cannot be retrained, is the proposed method still applicable?
>
> **A3:** As clearly demonstrated in Table 5 of our main paper, UAT provides a dual advantage: it not only significantly improves OOD detection performance but also enhances the classification accuracy of the ID datasets. Given these substantial performance benefits, there is little practical justification for not retraining the model, unless the ID training data is strictly inaccessible.
>
> If re-training is absolutely prohibited, the UAT component is indeed not applicable. In that specific scenario, the CCFN can still be used independently as a post-hoc refinement tool to optimize the OOD score of the existing model.
>
> **Q4:)** How can this method be extended beyond classification models such as object detection or semantic segmentation?
>
> **A4:** We thank the reviewer for this forward-looking question. The core principles of UAT and CCFN are directly extensible to dense prediction tasks.
>
> The mechanism for UAT is elegant: it does not require any auxiliary OOD loss or regularization loss. The goal of the UAM remains to maximize the task-specific loss (*L_{Task}*). For dense prediction, this involves replacing the classification loss *L_{CE}* with the task-specific total loss *L_{Task}* (e.g., combined classification and regression losses for detection). The UAM would apply universal regularization to the backbone features, forcing the network to learn robust region-level or pixel-level embeddings by maximizing *L_{Task}*.
>
> Similarly, CCFN can be incorporated into dense prediction architectures because these models also operate on multi-channel feature maps. The max-channel calibration mechanism can help detectors and segmenters downweight noisy activated channels and highlight those that better align with meaningful object or region semantics.

---

> ### Author Response · Authors · 2025-11-27
> **Follow-up on Rebuttal for Paper ID 6514**
>
> Dear reviewer roTf,
>
> We have addressed all the raised questions and concerns comprehensively. We hope that a quick review of our rebuttal will confirm that your valuable feedback has been thoroughly incorporated.
>
> If you have any remaining questions or require further clarification on our rebuttal or the revised paper content, please do not hesitate to reach out. If our revisions and explanations have resolved your concerns, we would be grateful for your support and encourage you to reflect this in your final score.
>
> Thank you.
> The authors

---

### Official Review · Reviewer_dj6D · 2025-11-01

**Soundness:** 2
**Presentation:** 3
**Contribution:** 2
**Rating:** 4
**Confidence:** 4

**Summary:**

This paper proposes a novel framework for OOD detection. The core idea is to actively enlarge the feature norm gap between ID and OOD data via two main contributions. First, the authors introduce Universal Adversarial Training (UAT), which learns a single, learnable Universal Adversarial Map (UAM) as a global regularizer to smooth decision boundaries and enhance the separability of ID classes. Second, to further enhance detection at inference, the paper proposes a Channel-Calibrated Feature Norm (CCFN) scoring mechanism, which purifies the feature norm signal by suppressing irrelevant background activations. Experimental results demonstrate that this method achieves significant performance improvements on several OOD detection benchmarks, including CIFAR-10/100 and ImageNet.

**Strengths:**

1. Unlike methods that passively rely on the feature norm gap, the proposed UAT framework actively enlarges this gap during training. The use of a single, learnable UAM as a global regularizer is an innovative point.
2. The experimental results are very strong. The proposed method (UAT + CCFN) consistently outperforms existing baselines on AUROC and FPR95 metrics across multiple backbones (ResNet, VGG, WRN) and datasets.
3. The paper provides a theoretical foundation for UAT, theoretically analyzing how UAT amplifies the norm ratio by enhancing intra-class compactness and inter-class separability.
4. The ablation study shows that UAT not only improves OOD detection but also enhances the classification accuracy of ID data.

**Weaknesses:**

1.  In Equation (2), the optimization objective of UAT is formulated as a max-min problem, which learns the universal adversarial map $L_{adv}$. However, the interaction between the cross-entropy loss and the adversarial perturbation in the objective function is not sufficiently explained, especially regarding how the perturbation alters the decision boundary. The method aims to regularize the model by pushing the decision boundary outwards, but the specific mechanism by which the adversarial map affects the decision boundary, particularly concerning class separation and feature compactness, is not fully articulated.
2.  The paper introduces a channel-wise feature norm recalibration method in Equation (5), where the feature norm is calculated by averaging the squared norms of each rescaled channel. While the idea of emphasizing channels with high activation values is intuitive, the paper does not explain why this specific norm aggregation method (i.e., averaging across channels) is the most effective. Specifically, using the maximum activation value ($w_{c}=max_{h,w}ReLU(f_{c,h,w})$) to rescale the contribution of each channel may not always capture the most discriminative features, especially in complex datasets where activation values might vary more subtly across spatial locations. An alternative approach, such as considering the variance or higher-order moments of activations, might provide a more robust recalibration. A deeper exploration of why this specific method was chosen and a comparison with other possible aggregation strategies would help clarify its effectiveness.
3.  The paper focuses on improving OOD detection performance but does not discuss the model's interpretability. Given the involvement of adversarial training, understanding how the universal adversarial map and channel recalibration affect the model's decision-making process would be beneficial.

**Questions:**

1. Can the authors provide a more in-depth analysis or visualization to specifically demonstrate how $L_{adv}$ (as described in Lemmas 1 and 2) leads to more compact intra-class features and more separated inter-class features?
2. The use of the maximum activation $w_{c}$ as a weight in CCFN is one choice. Did the authors try other aggregation statistics (e.g., mean, variance, or L2-norm) instead of $max$? How did their performance compare to the current method?
3. Can the authors provide some analysis on interpretability? For instance, do the patterns learned by the UAM ($L_{adv}$) help in identifying specific input features? The activation maps in Figure 3 show differences for CCFN on ID and OOD samples, but does this translate to an interpretable change in the model's decision logic?

---

> ### Author Response · Authors · 2025-11-24
> **We appreciate the reviewer’s(dj6D) insightful comments. Below, we address the questions raised by the reviewer. (Part 1)**
>
> # Reviewer dj6D
>
>
>
> **Q1:** In Equation (2), the optimization objective of UAT is formulated as a max--min problem that learns the universal adversarial map. However, the interaction between the cross-entropy loss and the adversarial perturbation is not sufficiently explained, particularly regarding how the perturbation alters the decision boundary. The method aims to regularize the model by pushing the decision boundary outward, but the mechanism by which the adversarial map affects the decision boundary, particularly concerning class separation and feature compactness, is not fully articulated. How do Lemmas 1 and 2 lead to more compact intra-class features and more separated inter-class features?
>
> **A1:** We thank the reviewer for raising this question. The interaction between the universal adversarial perturbation and the cross-entropy objective, and the resulting influence on the decision boundary, is discussed in the main paper (Sec. 3.1) and supported by the formal derivations in Appendix A.1 and A.2. In this rebuttal, we also added a visualization (Fig. 1 of the revised paper)that illustrates how the adversarial loss *L_adv* leads to more compact intra-class features and larger inter-class separation. Below, we summarize the intuition behind Lemmas 1 and 2.
>
> For intra-class compactness (Lemma 1), the requirement that the classifier remain robust against a universal perturbation encourages the model to smooth the decision boundary and push it outward from the data manifold. Geometrically, this forces the class-conditional feature distributions to become more convex and aligned. The optimization therefore drives features *f_θ(x)* to move closer to their corresponding class prototypes *W_k* in order to maintain consistent predictions under the perturbation. This results in reduced angular variance within each class (i.e., the angle between *f_θ(x)* and *W_k*), which corresponds to a tighter, more compact intra-class cluster.
>
> For inter-class separability (Lemma 2), the universal perturbation implicitly constrains the feature norms, so the model must enlarge the classification margin *z_k - z_j* through angular separation rather than feature scaling. As shown in Appendix A.2 (Eq. 12), the robust margin is directly proportional to the geometric term *(1 - \cos(β_{kj})*, where *β_{kj}* is the angle between two class prototypes *W_k* and *W_j*. Maximizing this margin, therefore, requires increasing the angular distance between prototypes. This angular expansion translates into greater Euclidean separation between class means and produces a more discriminative feature space under adversarial robustness.

---

> > ### Author Response · Authors · 2025-11-24
> > **We appreciate the reviewer’s(dj6D) insightful comments. Below, we address the questions raised by the reviewer. (Part 2)**
> >
> > **Q2:** The paper introduces a channel-wise feature norm recalibration method in Equation (5), where the feature norm is computed by averaging the squared norms of each rescaled channel. While emphasizing channels with high activation values is intuitive, the paper does not explain why this specific aggregation method is the most effective. Using the maximum activation as a rescaling factor may not always capture the most discriminative features. Alternative statistics (variance, higher-order moments) might provide a more robust recalibration. A deeper explanation of this choice and comparison with alternatives would be helpful.
> >
> > **A2:** We appreciate the reviewer's thoughtful question regarding the design choices in our channel-wise recalibration. Max-channel calibration is particularly effective because modern deep feature maps tend to contain a small subset of channels that exhibit strong, semantically meaningful activations [1]. Using the maximum activation as the rescaling factor amplifies these highly discriminative channels while naturally suppressing weak or noisy responses that often correspond to background content. Regarding the aggregation method, we average the squared norms of the rescaled channels because this quantity has been shown to approximate a confidence measure for the hidden classifier, as verified in [2]. This aggregation therefore provides a principled way to separate ID and OOD samples.
> >
> > To further support this choice, we evaluated several alternative calibration strategies in this rebuttal, including mean activation, minimum activation, variance, and the L2 norm of the feature map. The results in Table 1 show that the maximum-activation rescaling yields the strongest OOD detection performance. These empirical findings reinforce that the proposed max-channel calibration is not only intuitively aligned with the nature of deep feature representations but also empirically the most effective among the tested alternatives. For completeness, we have included this ablation study in Section A.10 of the revised manuscript
> >
> > | Method                | FPR↓ | AUC↑ |
> > | :-------------------- | :--: | :--: |
> > | No calibration        | 14.85 | 97.24 |
> > | L2 Norm calibration   | 14.63 | 97.35 |
> > | Variance calibration  | 14.97 | 97.31 |
> > | Mean calibration      | 22.26 | 95.76 |
> > | Min channel calibration | 99.96 | 22.26 |
> > | Max channel calibration | **12.21** | **97.80** |
> >
> > **Table 1:**: We evaluate the performance of various calibration strategies. The best results are highlighted in bold.
> >
> > **Q3:** The paper focuses on improving OOD detection performance but does not discuss interpretability. Given the involvement of adversarial training, understanding how the UAM and CCFN affect the model's decision-making process would be beneficial. Do the patterns learned by UAM help identify specific input features? The activation maps in Figure 3 show differences for ID and OOD samples, but does this translate to an interpretable change in the model's decision logic?
> >
> > **A3:** We appreciate the reviewer's interest in the interpretability aspect of our method. In this rebuttal, we provide an additional visualization (Fig. 1 of the revised paper) that illustrates how the UAM influences the decision boundary between classes. The theoretical basis for these effects is established in Lemmas 1 and 2, with detailed proofs in Appendices A1 and A2. As the model becomes robust to a universal perturbation, the intra-class features are pushed toward a more compact distribution, and the inter-class feature means become more separated. This adjustment in feature geometry also makes OOD samples less likely to be projected into ID regions, thereby directly improving OOD separability.
> >
> > The UAM, implemented through the adversarial objective L_adv, serves as a global regularizer that captures high-dimensional vulnerabilities shared across the ID manifold rather than targeting specific local input features. As shown in Figure 5 and further discussed in Appendix A9, the perturbation map itself resembles high-frequency noise and does not contain human-interpretable spatial patterns. Its interpretability therefore lies primarily in its indirect effects, namely how it reshapes the geometry of the representation space (Figure 3 in the main paper) and improves the model's generalization and robustness (Table 5 in the main paper).
> >
> > For CCFN, the activation maps in Figure 4 show a more directly interpretable effect. Channel calibration suppresses background or spurious activations and enhances responses in semantically meaningful regions. This behavior provides insight into which parts of the input most strongly influence the model's confidence and clarifies how the recalibrated features contribute to both ID and OOD decisions.

---

> > > ### Author Response · Authors · 2025-11-24
> > > **We appreciate the reviewer’s(dj6D) insightful comments. Below, we address the questions raised by the reviewer. (Part 3)**
> > >
> > > **Refernce:**
> > >
> > > [1] Jie Hu, Li Shen, and Gang Sun. Squeeze-and-excitation networks. In Proceedings of the IEEE conference on computer vision and pattern recognition, pp. 7132–7141, 2018.
> > >
> > > [2] Jaewoo Park, Jacky Chen Long Chai, Jaeho Yoon, and Andrew Beng Jin Teoh. Understanding the feature norm for out-of-distribution detection. In Proceedings of the IEEE/CVF International Conference on Computer Vision (ICCV), pp. 1557–1567, October 2023.

---

> ### Author Response · Authors · 2025-11-27
> **Follow-up on Rebuttal for Paper ID 6514**
>
> Dear reviewer dj6D,
>
> We have addressed all the raised questions and concerns comprehensively. We hope that a quick review of our rebuttal will confirm that your valuable feedback has been thoroughly incorporated.
>
> If you have any remaining questions or require further clarification on our rebuttal or the revised paper content, please do not hesitate to reach out. If our revisions and explanations have resolved your concerns, we would be grateful for your support and encourage you to reflect this in your final score.
>
> Thank you.
> The authors

---

### Official Review · Reviewer_LHzD · 2025-11-02

**Soundness:** 2
**Presentation:** 2
**Contribution:** 1
**Rating:** 2
**Confidence:** 4

**Summary:**

The paper proposes a  framework for  OOD detection by  enlarging the feature norm gap between in-distribution ID and OOD data. The core of the method is a Universal Adversarial Training  method, which uses a single, learnable Universal Adversarial Map  as a global regularizer. This UAT process regularizes the model's decision boundaries, which enlarges the gaps between ID classes and enhances generalization. Authors claim this improves ability to distinguish ID classes  translates into a superior ability to detect OOD samples, which are less likely to fall within the ID class boundaries. Then the authors introduce a Channel-Calibrated Feature Norm  scoring mechanism for test time. CCFN refines the feature norm by suppressing irrelevant background activations and emphasizing important channels, leading to a more accurate OOD score.

**Strengths:**

OOD detection is an important task specifically in real world setup.

The paper is generally well written and clear in its problem definition.

**Weaknesses:**

W1) Given that OOD detection in vision is a deployment-oriented task, I expect evaluation on larger, more challenging, real-world datasets. The experiments use CIFAR-10/100 and ImageNet as ID datasets; these are aging benchmarks for OOD detection. I expect consideration of application-driven datasets (e.g., medical imaging) and inclusion of comparisons to recent SOTA methods [E,F]. For instance, see [E], which reports ~100% AUC on CIFAR-10 vs. CIFAR-100. What does your work contribute over such baselines?


W2) How does the approach extend to ViT or other non-CNN backbones? The paper appears to report results only with CNNs.


W3) There are prior works at the intersection of adversarial perturbations and OOD detection [A,B,C]. The paper should cite and discuss these, and clearly articulate conceptual and algorithmic differences from the proposed framework. (I know they are designed specifically for robust OOD detection, but their frameworks share a lot of similarity.)


W4) The perturbation budget and norm used and the value of epsilon, are not sufficiently discussed.



[A] ATOM: Robustifying Out-of-distribution Detection Using Outlier Mining

[B] RODEO: Robust Outlier Detection via Exposing Adaptive Out-of-Distribution Samples

[C] Adversarially Robust Out-of-Distribution Detection Using Lyapunov-Stabilized Embeddings

[E] Deep Hybrid Models for Out-of-Distribution Detection

[F] Category-Extensible Out-of-Distribution Detection via Hierarchical Context Descriptions

**Questions:**

See Weaknesses.

---

> ### Author Response · Authors · 2025-11-24
> **We appreciate the reviewer’s(LHzD) insightful comments. Below, we address the questions raised by the reviewer(Part 1)**
>
> # Reviewer LHzD
>
>
> **Q1:** Given that OOD detection in vision is a deployment-oriented task, I expect evaluation on larger, more challenging, real-world datasets. The experiments use CIFAR-10/100 and ImageNet as ID datasets; these are aging benchmarks for OOD detection. How does the approach extend to ViT or other non-CNN backbones? The paper appears to report results only with CNNs.
>
> **A1:** We thank the reviewer for the suggestion. We have extended our method to a ViT backbone and evaluated its performance on the MIDOG [1] and SpaceNet-8 [2] datasets.
>
> -   We thank the reviewer for the suggestion. We have extended our method to a ViT backbone and evaluated its performance on the MIDOG [1] and SpaceNet-8 [2] datasets.The MIDOG (Mitosis Domain Generalization) dataset is a digital pathology dataset, focusing on mitotic figure detection in H&E-stained breast cancer tissue. The MIDOG dataset is divided into several sub-datasets, and each sub-dataset represents a distinct domain with variations in imaging conditions. The domains differ in scanner type, staining protocol, image resolution, acquisition center, and overall visual style. Specifically, datasets 1a-c are sourced from different scanners within the same institution, datasets 2-7 introduce differences in staining protocols, imaging devices, acquisition centers, tissue preparation, and different cancer types from human and canine species. In our evaluation, we used 1a-c as our ID, whereas the rest were used as OOD.
>
> -   SpaceNet-8 is a large-scale remote sensing dataset for flood detection, containing multi-spectral satellite imagery from several geographic regions. In our experiment on SpaceNet-8, we treat non-flooded images from two cities as ID, and use flooded images as OOD samples to evaluate the model's ability to detect distributional shift.
>
> #### Robustness across Backbone Architectures (ViT).
>
> To demonstrate the generality of our approach, we extended the UAT framework to the Vision Transformer (ViT) backbone. The core logic of UAT remains applicable, but we adapt the OOD scoring mechanism to the ViT's feature space, following established practices for OOD detection in ViT. Specifically, after training ViT with UAT, we extract the penultimate-layer features and apply a channel-wise gating mechanism (max activation) to emphasize the most informative dimensions for inference. To calculate the norm values, we follow the ViM [3] formulation. We compute the feature center using the pseudo-inverse of the classifier weights and estimate the covariance of centered features. The smallest-variance eigenvectors form an orthogonal subspace, and projecting features onto this subspace yields the orthogonal projection norm, which quantifies deviation from the ID feature manifold.
>
> **Experiment Setup.** To ensure a fair comparison, all baseline methods were run using their official repositories. Among these baselines, CATEX[4] is a CLIP-based approach pretrained on ImageNet. To adapt CATEX to MIDOG and SpaceNet-8, we followed its original design and constructed domain- and label-specific textual prompts, then finetuned the model to identify visual patterns that deviate from the expected distribution, such as unfamiliar cellular textures or image domains unrelated to the training data.
>
> **Results.** As shown in Tables 1 and 2, our method consistently yields substantial improvements over all competing approaches on both datasets. Notably, CATEX performs significantly worse than the other baselines, which we believe is due to the mismatch between the general-purpose semantics encoded in the ImageNet-pretrained CLIP backbone and the highly specialized, fine-grained visual cues required for medical pathology (MIDOG) and satellite imagery (SpaceNet-8). These results demonstrate that UAT remains effective across both CNN and non-CNN architectures and that it generalizes well to real-world distribution shifts that go far beyond conventional OOD detection benchmarks. We have added these new results to Section A.12 of the revised manuscript for completeness.

---

> ### Author Response · Authors · 2025-11-24
> **We appreciate the reviewer’s(LHzD) insightful comments. Below, we address the questions raised by the reviewer(Part 2)**
>
> | Method               | 2 FPR↓ | 2 AUC↑ | 3 FPR↓ | 3 AUC↑ | 4 FPR↓ | 4 AUC↑ | 5 FPR↓ | 5 AUC↑ | 6a FPR↓ | 6a AUC↑ | 6b FPR↓ | 6b AUC↑ | 7 FPR↓ | 7 AUC↑ | Avg FPR↓ | Avg AUC↑ |
> | :------------------- | :----: | :----: | :----: | :----: | :----: | :----: | :----: | :----: | :-----: | :-----: | :-----: | :-----: | :----: | :----: | :------: | :------: |
> | SCALE (ResNet50) [5]      | 67.60  | 72.53  | 67.51  | 72.43  | 28.20  | 85.78  | 54.84  | 78.85  |  66.09  |  69.06  |  69.88  |  78.29  | 55.24  | 78.15  |  58.48   |  76.44   |
> | OTOD (ResNet50) [6]       | 64.61  | 69.28  | 66.16  | 67.07  | 22.59  | 92.65  | 63.31  | 66.38  |  59.38  |  69.77  |  58.38  |  80.70  | 61.17  | 71.46  |  56.51   |  73.90   |
> | NAC-UE (ResNet50) [7]     | 59.08  | 75.91  | 59.61  | 77.80  | 30.14  | 82.46  | 56.23  | 71.79  |  57.99  |  76.61  |  61.88  | 80.96| 58.95  | 75.28  |  54.84   |  77.26   |
> | CATEX (CLIP) [4]         | 75.41  | 70.87  | 71.22  | 74.65  | 20.30  | 92.23  | 70.60  | 68.60  |  60.01  |  75.66  |  86.18  |  73.74  | 68.29  | 75.53  |  64.57   |  75.90   |
> | ViM (ViT) [3]           | 55.37| 77.14| **48.39** | 75.56  | 23.73  | 92.12  | **48.24** | 79.59  |  **41.04** | 83.39| 89.42  | 63.14  | 50.90| 76.25  |  51.01   |  78.17   |
> | Ours (ResNet50)        | 58.86  | 76.57  | 52.16  | 81.49| 18.91| 93.42| 48.52| 81.02| 48.32  | 79.49  | 67.91| 73.38  | **48.03** | **85.74** | 48.96| 80.56|
> | Ours (ViT)           | **51.20** | **81.29** | 53.17| **81.91** | **17.91** | **94.95** | 49.05  | **82.21** | 44.93| **85.80** | **60.14** | **80.54** | 53.89  | 80.50| **47.18** | **83.89** |
>
> **Table 1:**: OOD detection results on the MIDOG dataset. Because the MIDOG sub-datasets are indexed by numeric codes rather than descriptive names, we denote each OOD sub-dataset using its code in the first row. These sub-datasets differ from the ID data due to variations in staining protocols, imaging devices, acquisition centers, tissue preparation, and underlying cell types. The best results are highlighted in bold.
>
>
> | Method               | FPR↓ | AUC↑ |
> | :------------------- | :--: | :--: |
> | SCALE (ResNet50)       | 25.29 | 86.68 |
> | OTOD (ResNet50)        | 30.57 | 83.81 |
> | NAC-UE (ResNet50)      | 25.74 | 86.25 |
> | CATEX (CLIP)         | 29.33 | 82.19 |
> | ViM (ViT)            | 23.17| 90.18|
> | Ours (ResNet50)        | 24.63 | 88.46 |
> | Ours (ViT)           | **21.8** | **92.51** |
>
> **Table 2:**: OOD detection results on the SpaceNet-8 dataset. The non-flooded datasets from two cities are used as ID, whereas the flooded dataset is used as OOD. The best  are highlighted in bold.

---

> ### Author Response · Authors · 2025-11-24
> **We appreciate the reviewer’s(LHzD) insightful comments. Below, we address the questions raised by the reviewer(Part 3)**
>
> **Q2:** Inclusion of comparisons to recent SOTA methods [8], [4]. For instance, see [8], which reports 100% AUC on CIFAR-10 vs. CIFAR-100. What does your work contribute over such baselines?
>
> **A2:** We thank the reviewer for highlighting these recent baselines. In this rebuttal, we have included a direct comparison with [4], and the results are presented in Tables 1 and 2. As shown, CATEX performs noticeably worse than the other baselines on both MIDOG and SpaceNet-8. We attribute this to a mismatch between the broad, general-purpose semantics encoded by the ImageNet-pretrained CLIP backbone and the highly specialized, fine-grained visual characteristics required in medical pathology and satellite imagery.
>
> Regarding DHM [8], we have carefully examined this method and summarize our reasons for excluding it from our comparative analysis. A closer examination of DHM in [9] shows that its reported near-perfect OOD performance is largely a consequence of *sacrificing the model's fundamental classification ability*. The method requires assigning a substantially large weight to the flow likelihood loss in order to compress or partially collapse the ID feature space. This collapse artificially inflates the OOD score because ID representations become overly concentrated. Moreover, this effect occurs only within a very narrow and unstable range of loss weights. If the flow loss weight is increased slightly beyond this range, the encoder collapses not only the ID features but also the OOD features, eliminating any meaningful separation between them and causing the method to break down entirely. These observations indicate that DHM's strong OOD numbers do not result from genuinely improved distributional discrimination, but rather from a brittle configuration that exploits collapse behavior in the representation space. In contrast, our UAT improves ID classification and OOD detection simultaneously, suggesting that it enhances the underlying representation quality in a stable and principled way.
>
> Furthermore, the robustness of DHM's results is questionable. The revisiting study [9] showed that the original reported near-perfect AUC values (e.g., on CIFAR-10 vs. CIFAR-100) are not reproducible. The authors of the revisiting work found overlapping and ambiguous images across CIFAR-10 and CIFAR-100 (such as vans appearing in both 'car' and 'bus' classes). Given the presence of these overlapping samples, attaining a near-perfect OOD detection rate is theoretically impossible. Consistent with these observations, our re-implementation of DHM likewise did not achieve the near-perfect AUC values reported in the original work (see Table 10 of the revised paper). Therefore, we conclude that DHM's apparent performance is inflated by unstable shortcuts and cannot serve as a reliable baseline.

---

> ### Author Response · Authors · 2025-11-24
> **We appreciate the reviewer’s(LHzD) insightful comments. Below, we address the questions raised by the reviewer(Part 4)**
>
> **Q3:** There are prior works at the intersection of adversarial perturbations and OOD detection \[A, B, C\]. The paper should cite and discuss these, and clearly articulate conceptual and algorithmic differences from the proposed framework.
>
> **A3:** We thank the reviewer for suggesting the discussion of prior works \[A\] ATOM [10], \[B\] RODEO [11], and \[C\] AROS [12]. These papers study how to improve OOD detection by enhancing their adversarial robustness, but our proposed UAT + CCFN framework establishes a distinct and more efficient approach.
>
> The core difference lies in data dependency and computational strategy. ATOM and RODEO are both built upon the outlier exposure (OE) paradigm, meaning their robustness inherently relies on training with auxiliary OOD datasets [10] or synthetically generated OOD samples using a pre-trained diffusion model [11]. Our UAT framework, however, is designed for the stricter setting where we achieve superior OOD separation solely by learning from the ID data, which is a crucial advantage when external OOD data is unavailable.
>
> Furthermore, these three related works typically rely on per-instance adversarial perturbations for training and scoring. Our UAT framework, in contrast, optimizes a single, learnable UAM. This approach is significantly more efficient because the UAM is simply added directly to the test input, eliminating the need to generate a new, complex perturbation for each test sample during inference.
>
> Regarding Paper \[C\] AROS [12], which uses Lyapunov-Stabilized Embeddings, this method requires specialized architectures, potentially neural ordinary differential equations, to model the feature space as a dynamic system. This dependency on differential equation solvers leads to a substantial increase in computational complexity, resulting in significantly slower training and inference speeds compared to our UAT method, which is built upon standard backbone architectures (e.g., ResNet, WRN). In conclusion, our UAT+CCFN offers a solution that is both conceptually distinct from OE-based methods and computationally superior to per-instance and specialized architectural approaches.We have added the discussion related to these papers in the related work section Lines 121-130
>
> **Q4:** The perturbation budget and norm used, and the value of epsilon, are not sufficiently discussed.
>
> **A4:** We appreciate the reviewer's question regarding the perturbation magnitude. In our method, the perturbation map is not controlled through a fixed budget epsilon. Instead, the learnable perturbation is scaled by a small coefficient α, which we set to 0.1, and the resulting perturbed inputs are clipped to the valid image range [0,1]. These two mechanisms work together to keep the perturbation magnitude naturally bounded and to avoid training instability. We would also like to note that the influence of this coefficient was already analyzed in the original submission, specifically in Appendix A8 and Table 12, where the method remains stable across different choices of α. Furthermore, the methodology and the effect of the perturbation norm on OOD detection are explicitly analyzed in Sections 3.3 and 5.4.

---

> > ### Author Response · Authors · 2025-11-24
> > **We appreciate the reviewer’s(LHzD) insightful comments. Below, we address the questions raised by the reviewer(Part 5)**
> >
> > **Refernce:**
> >
> > [1] Marc Aubreville, Frauke Wilm, Nikolas Stathonikos, Katharina Breininger, Taryn A Donovan, Samir Jabari, Mitko Veta, Jonathan Ganz, Jonas Ammeling, Paul J van Diest, et al. A comprehensive multi-domain dataset for mitotic figure detection. Scientific data, 10(1):484, 2023.
> >
> > [2] Ronny Hänsch, Jacob Arndt, Dalton Lunga, Matthew Gibb, Tyler Pedelose, Arnold Boedihardjo, Desiree Petrie, and Todd M Bacastow. Spacenet 8-the detection of flooded roads and buildings. In Proceedings of the IEEE/CVF conference on computer vision and pattern recognition, pp. 1472–1480, 2022.
> >
> > [3] Xiaoyun Wang, Weitang Liu, Shiyu Liang, and Yixuan Li. Vim: Out-of-distribution with virtual-logit matching. In Proceedings of the IEEE/CVF Conference on Computer Vision and Pattern Recognition, pp. 4921–4930, 2022.
> >
> > [4] Kai Liu, Zhihang Fu, Chao Chen, Sheng Jin, Ze Chen, Mingyuan Tao, Rongxin Jiang, and Jieping Ye. Category-extensible out-of-distribution detection via hierarchical context descriptions. Advances in Neural Information Processing Systems, 36:33241–33261, 2023a.
> >
> > [5] Kai Xu, Rongyu Chen, Gianni Franchi, and Angela Yao. Scaling for training time and post-hoc out-of-distribution detection enhancement. arXiv preprint arXiv:2310.00227, 2023.
> >
> > [6] Heng Gao, Zhuolin He, and Jian Pu. Detecting ood samples via optimal transport scoring function. In ICASSP 2025-2025 IEEE International Conference on Acoustics, Speech and Signal Processing (ICASSP), pp. 1–5. IEEE, 2025.
> >
> > [7] Yibing Liu, Chris Xing Tian, Haoliang Li, Lei Ma, and Shiqi Wang. Neuron activation coverage: Rethinking out-of-distribution detection and generalization. arXiv preprint arXiv:2306.02879, 2023b.
> >
> > [8] Senqi Cao and Zhongfei Zhang. Deep hybrid models for out-of-distribution detection. In Proceedings of the IEEE/CVF Conference on Computer Vision and Pattern Recognition, pp. 4733–4743, 2022.
> >
> > [9] Paul-Ruben Schlumbom and Eibe Frank. Revisiting deep hybrid models for out-of-distribution detection. 2025.
> >
> > [10] Tianlong Chen, Zhenyu Zhang, Zhen Ding, Yang Liu, and Zhangyang Wang. Atom: A general framework for out-of-distribution detection using outlier mining. In AAAI, 2021.
> >
> >
> > [11] Hossein Mirzaei, Mohammad Jafari, Hamid Reza Dehbashi, Ali Ansari, Sepehr Ghobadi, Masoud Hadi, Arshia Soltani Moakhar, Mohammad Azizmalayeri, Mahdieh Soleymani Baghshah, and Mohammad Hossein Rohban. Rodeo: Robust outlier detection via exposing adaptive out-of-distribution samples. arXiv preprint arXiv:2501.16971, 2025.
> >
> >
> > [12] Hossein Mirzaei and Mackenzie W Mathis. Adversarially robust out-of-distribution detection using lyapunov-stabilized embeddings. arXiv preprint arXiv:2410.10744, 2024.

---

> ### Author Response · Authors · 2025-11-27
> **Follow-up on Rebuttal for Paper ID 6514**
>
> Dear reviewer LHzD,
>
> We substantially expanded our empirical evidence by integrating new experimental results on two challenging, real-world application datasets, MIDOG and SpaceNet-8. Furthermore, we successfully tested our UAT framework on the modern Vision Transformer (ViT) architecture. The results consistently validate the effectiveness of UAT for superior OOD detection across diverse domains and model families.
>
> If you have any remaining questions or require further clarification on our rebuttal or the revised paper content, please do not hesitate to reach out. If our revisions and explanations have resolved your concerns, we would be grateful for your support and encourage you to reflect this in your final score.
>
> Thank you.
> The authors

---

### Author Response · Authors · 2025-11-26
**Summary of Revisions and Key Updates for Our Rebuttal**

Dear AC and Reviewers,

We appreciate the time you have taken to assess our paper. This note summarizes our rebuttal, which addresses all raised questions and concerns comprehensively. Our responses are direct and focused. Below, we highlight two major updates that significantly strengthen our submission:


**1. Expanded Empirical Validation**

We substantially broadened our empirical evidence by adding new experimental results on two challenging, real-world datasets **MIDOG** and **SpaceNet-8**. We also tested our UAT framework on the modern **Vision Transformer (ViT)** architecture. These results consistently validate the effectiveness of UAT for robust OOD detection across diverse domains and model families. (Reviewer LHzD)



**2. Enhanced Theoretical Clarity**

In addition to the existing theoretical foundation, we added an **illustrative example (new Figure 1 in the revised manuscript)** to visually explain how UAT smooths and modifies the decision boundary. This directly supports user understanding of the geometric effect central to our method. (Reviewer dj6D and Reviewer Zdr8)



We also addressed several notable concerns:

-  We explained that the perfect score reported in the DHM [1] paper is not robust but results from collapse behavior in the representation. This observation is supported by a recent publication [2]. (Reviewer LHzD)
-  We expanded the related-work section and included the prior works indicated by Reviewer LHzD.
-  We added an ablation study with different aggregation strategies for the proposed CCFN scoring mechanism, as suggested by (Reviewer dj6D).
-  We included training and testing time statistics for ResNet-50 on ImageNet. (Reviewer roTf)
-  We updated the evaluation table to include the prior works noted by (Reviewer LHzD and Reviewer Zdr8).



We believe these additions, together with our detailed responses, fully address the concerns raised. We hope a quick review of the rebuttal confirms that your valuable feedback has been thoroughly incorporated.

Sincerely,
The Authors

---

### References
[1] Senqi Cao and Zhongfei Zhang. *Deep hybrid models for out-of-distribution detection*. CVPR 2022.
[2] Schlumbom, P.-R., & Frank, E. (2025). *Revisiting Deep Hybrid Models for Out-of-Distribution Detection*. TMLR.

---

### Note · Authors · 2026-01-20

I have read and agree with the venue's withdrawal policy on behalf of myself and my co-authors.